# SPRINT: Enabling Interleaved Planning and Parallelized Execution in Reasoning Models

Emil Biju[1,2]* Shayan Talaei[1]* Zhemin Huang[1]* Mohammadreza Pourreza[3]

Azalia Mirhoseini[1]† Amin Saberi[1]†

[1]Stanford University    [2]Microsoft    [3]Google

{emilbiju, stalaei, zheminh}@stanford.edu
pourreza@google.com, {azalia, saberi}@stanford.edu

## Abstract

Large reasoning models (LRMs) excel at complex reasoning tasks but typically generate lengthy sequential chains-of-thought, resulting in long inference times before arriving at the final answer. To address this challenge, we introduce SPRINT, a novel post-training and inference-time framework designed to enable LRMs to dynamically identify and exploit opportunities for parallelization during their reasoning process. SPRINT incorporates an innovative data curation pipeline that reorganizes natural language reasoning trajectories into structured rounds of long-horizon planning and parallel execution. By fine-tuning LRMs on a small amount of such curated data, the models *learn* to dynamically identify independent sub-tasks within extended reasoning processes and effectively execute them in parallel. Through extensive evaluations, we demonstrate that models fine-tuned with the SPRINT framework match the performance of reasoning models on complex domains such as mathematics while generating up to 39% fewer sequential tokens on problems requiring more than 8,000 output tokens. Finally, we observe consistent results transferred to two out-of-distribution tasks, namely GPQA and Countdown, with up to 45% and 65% reduction in average sequential tokens respectively for longer reasoning trajectories, while matching the performance of the fine-tuned reasoning model.

## 1 Introduction

Scaling inference-time compute in large language models (LLMs) has consistently been shown to enhance reasoning accuracy. Existing methods broadly fall into two categories: sequential [1] and parallel [2]. Sequential approaches, notably large reasoning models (LRMs) such as Deepseek-R1 [3] and OpenAI o1 [4], have demonstrated remarkable successes in solving complex reasoning tasks, e.g., math and coding, but at the cost of generating very lengthy sequences of tokens. On the other hand, parallel methods, such as repeated sampling with self-consistency [5] or best-of-N [6, 7] leverage multiple response generations to improve accuracy. However, these methods typically lack effective coordination and shared information across inference paths, leading to redundant computations and limited performance gains. Furthermore, structured parallel methods like Tree-of-Thoughts [8] and Graph-of-Thoughts [9] require predefined, heuristics-driven search structures, inherently restricting flexibility and scalability across diverse tasks.

---

*Equal contribution.

†Equal senior authorship.

39th Conference on Neural Information Processing Systems (NeurIPS 2025).

We propose **SPRINT**[3], a framework for post-training and inference of reasoning models that combines the advantages of sequential reasoning and parallel inference, while maintaining the flexibility required for general tasks. Instead of relying on manual structures, SPRINT trains reasoning language models to dynamically identify and exploit parallelization opportunities during inference. This enables SPRINT to achieve the high accuracy of reasoning models while significantly reducing the number of sequential tokens needed for solving complex reasoning tasks such as mathematics.

For the inference, SPRINT introduces an orchestration of LRMs through two distinct roles: a planner and a pool of executors. At each step, the planner that has access to the cumulative context of the reasoning trajectory generates a set of independent plans, each explained via a natural language *<prompt>*. Subsequently, multiple executors concurrently carry out these plans. This interleaved planning-execution strategy accelerates the reasoning process by enabling simultaneous execution of lengthy tasks.

Although many off-the-shelf LRMs achieve high performance via sequential reasoning trajectories, they are not trained for effectively proposing parallelizable tasks. Recognizing that LRMs' reasoning trajectories for a given query include steps such as reflection on their previous steps, decomposing tasks to subtasks, and trial-and-error exploration of alternative strategies, we question the necessity of strictly sequential reasoning. In practice, many reasoning steps are independent and thus can be executed in parallel; for instance, by simultaneously exploring multiple strategies or independently computing separate components of a complex problem. Building on these insights, we designed a data curation pipeline that carefully reorganizes natural language reasoning trajectories into structured plans and parallel executions, closely preserving the original data distribution. Finally, through supervised fine-tuning of the reasoning model on only 1700 such demonstrations, we unlock the model's capability to dynamically recognize and exploit opportunities for parallel reasoning.

To evaluate the accuracy and efficacy of SPRINT, we conducted experiments on MATH-500 [7] for testing in-distribution, and two out-of-domain distribution benchmarks: GPQA-diamond [10], and Countdown (Game of 24) [8]. On MATH-500, SPRINT improved the accuracy of the base reasoning model Deepseek-R1-distill-7B [3] from 89.1% to 92.5%, outperforming the reasoning fine-tuned model (RFT) at 91%, while generating 440 fewer sequential tokens on average. On the problems requiring longer reasoning trajectories (more than 8000 tokens under the RFT model), SPRINT achieves even greater savings, reducing sequential tokens by up to 39%. We also show that SPRINT generalizes well to out-of-domain tasks, matching the performance of the reasoning fine-tuned model while significantly reducing token usage – by 53% on Countdown.

In summary, our work makes the following key contributions[4]:

- We propose SPRINT, an innovative framework for accelerating the reasoning process of large reasoning models through rolling horizon parallel planning and execution.

- We develop a novel data curation pipeline that carefully converts complex natural language reasoning trajectories into structured datasets for fine-tuning LRMs, featuring a multi-step process that includes step extraction, Directed Acyclic Graph (DAG) creation, packing, filtering, and reformatting.

- We analyze the accuracy and the efficiency of SPRINT on complex reasoning tasks in comparison to strong reasoning baselines. Our results show that SPRINT can achieve higher accuracy compared to the reasoning distilled model, while generating up to 39% fewer sequential tokens on long reasoning trajectories.

- We show consistent generalization performance of SPRINT on two out-of-domain benchmarks, saving sequential tokens by about 45% on GPQA and 65% on Countdown respectively, while matching the performance of the reasoning finetuned model. These results highlight SPRINT's ability to effectively parallelize reasoning trajectories across diverse domains.

---

[3]The name SPRINT is inspired by the agile development methodology, where a sprint involves a planning phase followed by parallel, incremental execution.

[4]We open-source our code and datasets at this repository.

Table 1: Comparison of inference-time scaling approaches. Methods are evaluated based on support for inference-time parallelism, adaptive search, model optimization, and the capability to handle multi-step sequential reasoning. SPRINT uniquely addresses all criteria, enabling dynamic parallelism in general reasoning tasks that require interdependent sequential steps.

| Method | Inference-Time Parallelism | Adaptive Search | Model Optimization | Multi-Step Reasoning |
|---|:---:|:---:|:---:|:---:|
| Tree-of-Thought (ToT) [8] | ✓ | ✗ | ✗ | ✗ |
| Graph-of-Thought (GoT) [9] | ✓ | ✗ | ✗ | ✗ |
| Skeleton-of-Thought (SoT) [25] | ✓ | ✓ | ✗ | ✗ |
| Repeated Sampling [2, 5, 6] | ✓ | ✗ | ✗ | ✗ |
| Reasoning Models [3, 4] | ✗ | ✓ | ✓ | ✓ |
| PASTA [26] | ✓ | ✓ | ✓ | ✗ |
| Hogwild! Inference [27] | ✓ | ✓ | ✗ | ✗ |
| **SPRINT (Ours)** | ✓ | ✓ | ✓ | ✓ |

## 2 Related Work

**Long Chains-of-Thought for Improved Reasoning.** Recent advancements have shown that generating extensive chains-of-thought [1] significantly enhances the reasoning capabilities of large language models, particularly in tasks such as mathematical problem-solving and logical inference [11, 12, 4, 3]. Despite their effectiveness, these methods inherently produce long sequential outputs, increasing latency and slowing inference speed. SPRINT addresses this limitation by enabling models to dynamically parallelize independent reasoning steps, significantly reducing sequential generation and enhancing inference efficiency.

**Structured Search and Multi-Agent Frameworks.** Approaches like Tree-of-Thought [8], Graph-of-Thought [9], Forest-of-Thought [13], and Atom-of-Thought [14], along with multi-agent interaction methods [15, 16, 17, 18], structure reasoning processes through fixed search patterns or predefined interaction protocols, often at the full-solution level. SPRINT generalizes these frameworks by *training* models to autonomously allocate inference-time computation between serial and parallel tasks to solve sub-parts of one solution trajectory or explore alternative solutions.

**Planning and Execution with Language Models.** Integrating planning capabilities into language models has been explored through upfront decomposition of tasks into subtasks [19, 20, 21, 22] or iterative refinement based on intermediate feedback [23, 24]. These approaches primarily rely on sequential execution without explicitly considering dynamic parallel planning. SPRINT addresses this gap by enabling models to autonomously perform dynamic parallel planning, enhancing inference efficiency through concurrent execution.

**Parallelization in language model reasoning.** Methods that leverage parallel inference paths, such as best-of-N sampling [6, 7] or self-consistency [5], have shown performance improvements through generating multiple independent reasoning trajectories. However, these techniques typically lack effective coordination among parallel threads, resulting in redundancy and inefficient computation. To mitigate this issue, Skeleton-of-Thought (SoT)[25] and APAR[28] parallelize decoding by assuming semantic independence among subtasks, thus enabling separate processing of different response segments. Although these methods achieve faster inference, they exhibit suboptimal performance on tasks that inherently require sequential reasoning, such as mathematical problem-solving, where later steps depend on earlier computations.

Recently, three works, PASTA [26], Hogwild! Inference [27], and APR [29] have investigated parallelization within a shared reasoning trajectory. PASTA teaches models to decompose a task into parallel subtasks and subsequently merges their full context back into a single main thread, but it does not optimize for reasoning tasks that require multi-step planning. Hogwild! Inference relies on parallel prompting for collaborative reasoning among multiple workers, without tuning the models to distribute tasks effectively. APR trains models to delegate subtasks to parallel child threads for synthetic countdown tasks, but its training data curation relies on a *specialized symbolic solver*, limiting its applicability to general reasoning tasks. SPRINT extends this line of research by

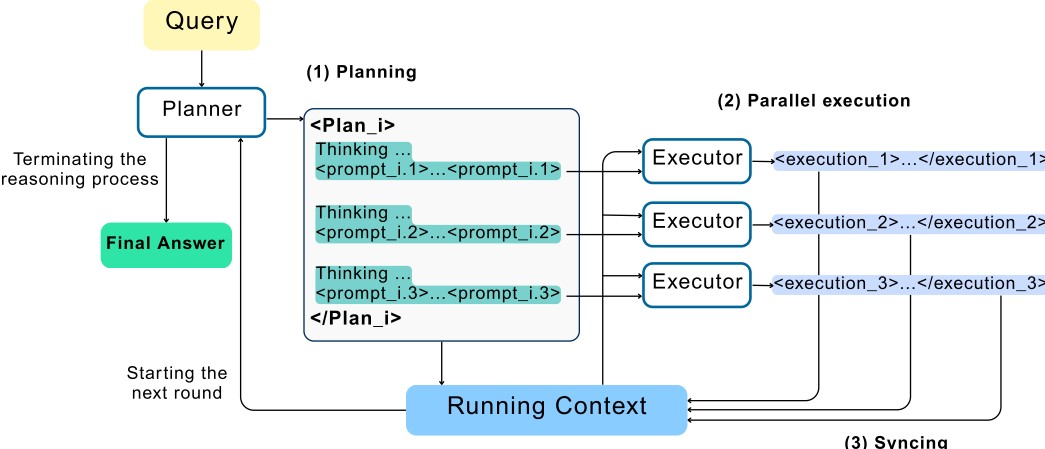

Figure 1: Overview of SPRINT's inference process: 1) The planner receives the cumulative context, including previous plans and execution results, and either proposes a new set of independent tasks or terminates the process by producing the final answer. 2) A pool of executors concurrently performs each task according to their prompts. 3) The execution outcomes are appended back into the cumulative context with corresponding tags, returning to step 1 for the next iteration.

introducing a generalizable post-training framework that enables reasoning models to dynamically structure inference for general reasoning tasks.

In general, an effective reasoning system should support logical multi-step interdependencies (multi-step reasoning) to accurately handle tasks where later steps depend on earlier outcomes. It should dynamically adapt its search strategy (adaptive search) to address diverse problem structures. Optimizing model performance specifically for downstream tasks (model optimization) is often necessary to achieve efficient results. Finally, leveraging parallel execution (inference-time parallelism) is crucial to reducing latency by concurrently processing independent reasoning subtasks. Table 1 compares our method and existing inference-time scaling methods against these criteria.

## 3 Methodology

In this section, we outline the design and components of SPRINT, which at a high level consists of an inference framework for reasoning models and a training protocol to teach them how to effectively identify and exploit parallelizable planning and execution during their reasoning processes.

### 3.1 Interleaved Planning and Parallel Execution at Inference Time

SPRINT's inference comprises two main modules: a planner and a pool of executors, all powered by fine-tuned reasoning models. Inference begins when the planner receives the *input query*, followed by iterative rounds of planning and execution, called *stages*, until the planner decides to terminate the process by producing the final answer. As shown in Figure 1, each inference stage includes the following three phases:

**1. Planning.** At stage $i$, the planner receives the cumulative context of the reasoning trajectory, which includes the input query, previous plans, and the execution outputs from all the preceding stages (1 through $i-1$). The planner then generates a plan for the current stage, enclosed within *<Plan_i>* tags. During this stage, the planner may generate intermediate reasoning tokens, benefiting from its reasoning capabilities. When the planner identifies a subtask suitable for delegation to an executor, it specifies this task within tags *<prompt_i.j>*. Upon closing each *</prompt_i.j>* tag, an executor initiates the corresponding task given the current cumulative context snapshot.

**2. Parallel executions.** Each executor independently and concurrently performs its assigned subtask by generating a chain-of-thought reasoning trajectory to accomplish the specific task. Executing these

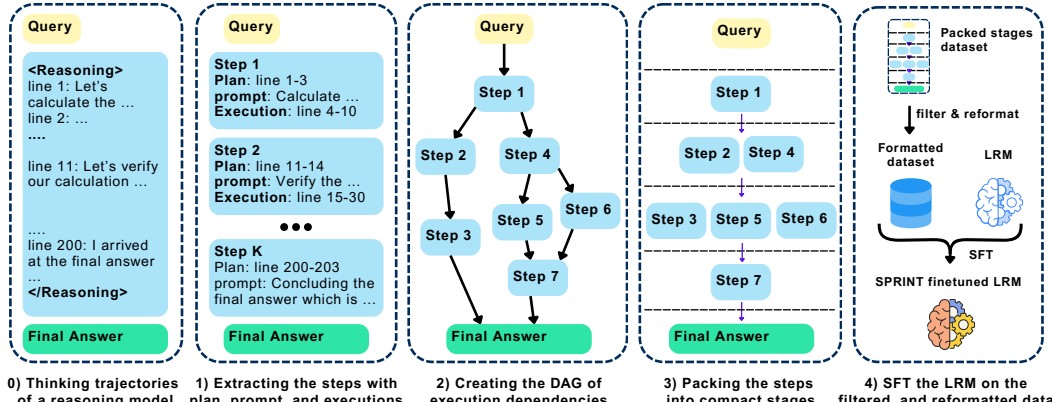

**0) Thinking trajectories of a reasoning model**    **1) Extracting the steps with plan, prompt, and executions**    **2) Creating the DAG of execution dependencies**    **3) Packing the steps into compact stages**    **4) SFT the LRM on the filtered, and reformatted data**

Figure 2: Overview of the SPRINT training pipeline: (0) Starting from raw reasoning trajectories, (1) we first extract individual reasoning steps, identifying their planning and execution phases. Next, (2) we construct a DAG representing dependencies among these steps, and then (3) group steps into compact stages that can be executed in parallel. Finally, (4) after filtering and reformatting these structured stages into training samples, we perform supervised fine-tuning of a reasoning model to dynamically propose and execute parallelizable tasks.

subtasks in parallel significantly reduces the total number of sequential tokens generated compared to processing them sequentially, greatly improving inference efficiency.

**3. Syncing.** Once all parallel executions are complete, the results from each executor are enclosed within tags *<execution_i.j>*, clearly indicating their corresponding tasks. These results are synced back into the cumulative context in the same order as their original prompt definitions. The updated context is then fed back to the planner, which either initiates the next stage or concludes the inference by outputting the final answer.

### 3.2 Training Reasoning Models for SPRINT Framework

To effectively train reasoning models to identify and exploit parallelization opportunities during inference, we developed a data curation pipeline that transforms complete natural language reasoning trajectories into structured rounds of rolling-horizon planning and parallel execution. The pipeline extracts individual planning and execution steps, organizes them into dependency-based stages, and generates training examples that capture both sequential planning and parallel execution aspects. An overview of this pipeline is shown in Figure 2. Detailed prompts for each step in the pipeline are provided in Appendix A.

**1. Step extraction.** Given a reasoning trajectory $\tau$, generated by DeepSeek-R1 [3] in response to a query $Q$, we decompose it into distinct steps $S = \{S_1, S_2, \ldots, S_n\}$ by prompting an LLM (in this case, GPT-4o) with specific instructions; refer to Appendix A.2. Each step $S_i$ is further decomposed into a planning phase ($P_i$), where R1 identifies tasks and strategies, and an execution phase ($E_i$), where these planned tasks are performed. Note that some steps may only involve planning without explicit execution; these are termed *plan-only steps*, and no executor instructions are generated for them.

To discourage trivial executor calls, we merge very short executions back into their planning phase, making them plan-only steps and encouraging the planner to handle simpler tasks independently.

**2. DAG creation.** Next, we identify dependencies among steps by prompting a smaller LLM (GPT-4o-mini) to determine which steps depend on others; for the instructions see A.2. These dependencies are represented formally as:

$$D = \{(S_i, S_j) \mid S_j \text{ depends on } S_i, i < j, S_i, S_j \in S\}.$$

This set of dependencies forms a Directed Acyclic Graph (DAG), denoted by $G = (S, D)$, where nodes represent individual steps and edges represent dependencies among them.

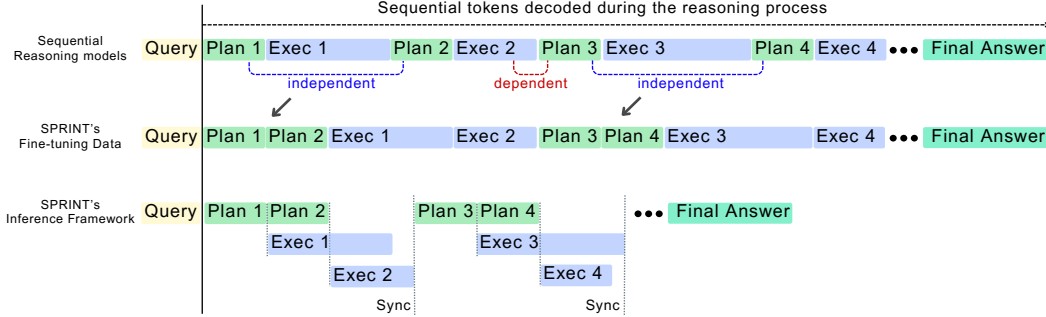

Figure 3: Comparison of sequential tokens decoded during reasoning. Sequential reasoning models generate all the steps serially, resulting in long token sequences. SPRINT's fine-tuning data restructures these steps into stages, grouping parallelizable plans followed by their respective executions. This organization enables SPRINT's inference framework to execute these grouped steps in parallel, significantly reducing the number of sequential tokens.

**3. Packing.** We group the steps into stages, each containing plans that can be generated simultaneously by the planner and executions that can be carried out concurrently by executors. While a naive approach would group steps solely based on their depth in the DAG, we further optimize the stage arrangement by observing that if the parent $S_p$ of a node $S_i$ is a plan-only step, $S_i$ can safely be included in the same stage as $S_p$. This optimization ensures both context availability and enhanced parallelization efficiency. Further details on this adjustment are provided in Appendix A.2.

Formally, the stage number $\sigma(S_i)$ for each step $S_i = (P_i, E_i)$ is defined as:

$$\sigma(S_i) = \begin{cases} 1, & \text{if } S_i \text{ has no parents} \\ \max_{S_p \in \text{Parents}(S_i)} \left(\sigma(S_p) + \mathbb{1}(E_p \neq \emptyset)\right), & \text{otherwise} \end{cases}$$

The set of steps at a given stage $k$ consists of all steps with stage number $\sigma(S_i) = k$, represented as:

$$\mathcal{L}^{(k)} = S_i \in S \mid \sigma(S_i) = k.$$

Within each stage $k$, the combined plan is created by concatenating the plans of all steps $S_i$ in $\mathcal{L}^{(k)}$, ordered according to their original sequence. The execution phase for stage $k$ includes execution components from all steps, excluding those that are plan-only:

$$\mathcal{P}^{(k)} = \text{concat}(P_i \mid S_i \in \mathcal{L}^{(k)}), \quad \mathcal{E}^{(k)} = \{E_i \mid S_i \in \mathcal{L}^{(k)}, E_i \neq \emptyset\},$$

where $E_i = \emptyset$ indicates that $S_i$ is a plan-only step.

**4. Training the LRM.** To ensure that the model learns from trajectories with significant parallelization potential, we introduce a *parallelization ratio*, defined as $(\#\text{steps})/(\#\text{stages})$, and discard trajectories with ratios below 1.5. The selected trajectories are reformatted into sequences of stage-wise plans and executions, enclosed within explicit tags (*<Plan_i>* and *<execution_i.j>*) in the order illustrated in Figure 3. Finally, we fine-tune the LRM on the reformatted thinking patterns. Through this process, the model learns to dynamically propose independent, parallelizable tasks based on previous sequences of plans and executions, and to execute each task following its corresponding prompt effectively.

**Methodology Overview.** Overall, as detailed in Section 3.2, SPRINT trains reasoning models to propose parallelizable subtasks rather than generating their entire reasoning trajectories serially. During inference, as described in Section 3.1, the trained model effectively manages long-term interdependencies while significantly reducing the number of sequential tokens generated. Figure 3 illustrates this workflow, highlighting how SPRINT reorganizes sequential reasoning traces into parallelizable stages during training and subsequently leverages this learned parallel structure for efficient, concurrent execution at inference time. For examples of SPRINT's reasoning versus serial reasoning trajectories, please refer to Appendix B.

# 4 Experiments

## 4.1 Experimental Setup

**Datasets.** To train our models, we begin with 6,000 reasoning trajectories from DeepSeek-R1 [3] generated on the training set of the MATH dataset [30], as released by [31]. After filtering these trajectories for correctness of the final answers and processing them through our data curation pipeline (Section 3.2), we obtain a curated set of approximately 1,700 samples for training.

For evaluation, we primarily use the MATH-500 benchmark [32], a widely recognized test set consisting of 500 mathematical reasoning problems. To further examine the generalization capabilities of SPRINT to more challenging and out-of-distribution scenarios, we evaluate its performance against strong baseline models on two additional benchmarks. First, we evaluate on GPQA-diamond [10], a dataset from entirely different scientific domains, including biology, physics, and chemistry, thus assessing cross-domain reasoning robustness. Moreover, following [29, 8], we test SPRINT on a subset of 1000 samples from Countdown [8], a synthetic numerical reasoning task in which models must derive a target number from four provided numbers using arithmetic operations $(+, -, \times, \div)$.

**Baselines.** We compare SPRINT against several reasoning baselines employing both serial and parallel sampling strategies:

**1. Base reasoning model (DeepSeek-R1-Distill-Qwen-7B)** [3]: This model is a distillation of the main R1 reasoning model into Qwen-2.5-7B [33], released by DeepSeek. We use this reasoning model both as a baseline for direct comparison and as the base model for our fine-tuning experiments.

**2. Reasoning fine-tuned model (RFT)**: To control for the effect of the training data and compare against conventional distillation methods, we perform supervised fine-tuning of the DeepSeek-R1-Distill-Qwen-7B model using the same 1,700 R1 reasoning trajectories from MATH used to train SPRINT. This model represents a standard continued distillation of Qwen-2.5-7B on R1 trajectories from the MATH dataset.

**3. Skeleton-of-Thought (SoT)** [25]: Given a query, SoT decomposes it into subtasks and executes them through parallel LLM calls within a single stage. Both the subtask generation and execution processes rely on out-of-the-box LLMs without any task-specific fine-tuning. We evaluate SoT using both the chat-instruct Qwen-2.5-7B model (referred to as *SoT-chat*) and the reasoning-focused DeepSeek-R1-Distill-Qwen-7B model (referred to as *SoT-reasoning*).

**4. Repeated Sampling + Self-consistency** [2, 5]: We include repeated sampling combined with self-consistency aggregation as a baseline to evaluate whether a purely parallel sampling approach can achieve similar accuracy and efficiency compared to the interleaved planning and execution framework of SPRINT.

**Evaluation Metrics.** We consider two metrics to evaluate the performance and efficiency of different approaches. First, we measure the **accuracy** of the final answer reached for the downstream task, computed as the percentage of the correctly answered queries by each method (see A.4 for details). Second, to evaluate the efficiency improvements in terms of the latency, we measure the number of **sequential tokens** generated by each method. In particular, for sequential reasoning baselines, it is exactly the number of output tokens. For SPRINT, we calculate the sequential tokens as follows:

$$\text{number of sequential tokens} = \sum_{i=1}^{\#\text{ stages}} \max_{k}^{\#\text{ prompts at stage } i} (P_{i.k} + E_{i.k}),$$

where $P_{i.k}$ and $E_{i.k}$ represent the number of sequential tokens generated by the planner until the end of $k^{\text{th}}$ prompt and by an executor for the $k^{\text{th}}$ execution at step $i$ respectively. Note that the ideal wall-clock time correlates with the number of sequential tokens generated by each method; however, accurately measuring this metric would require higher computational resources, which we discuss further in Section 5.

## 4.2 Results

**Comparison to conventional distillation.** Figure 4 shows the accuracy and average number of sequential tokens generated by different methods on the MATH-500 benchmark. We observe that fine-tuning our base model (R1-Distill-7B) on trajectories generated by DeepSeek-R1 improves

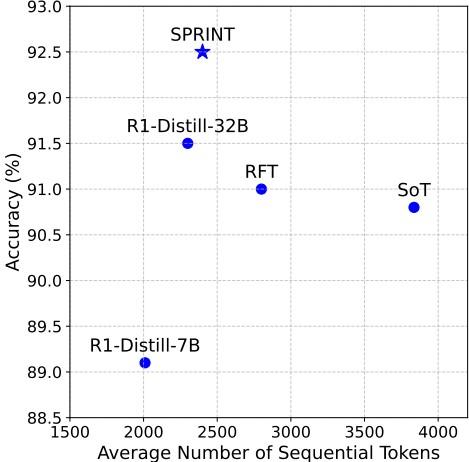
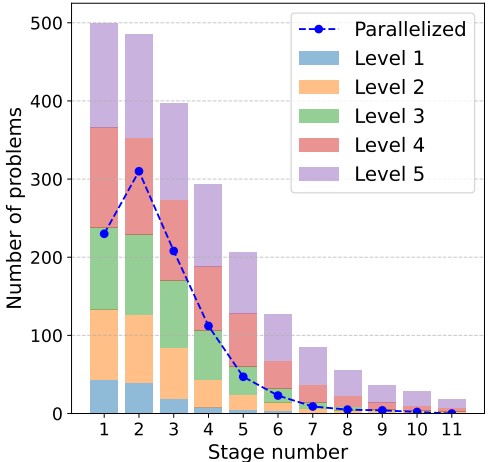

Figure 4: Pareto plot comparing accuracy (%) and sequential token counts generated by different methods on MATH-500. While SPRINT achieves slightly higher accuracy compared to the RFT model, it generates 440 ($\sim 15\%$) fewer tokens on average.

Figure 5: Number of problems at each difficulty level in MATH-500 that pass each stage of interleaved planning before arriving at the final answer. The dashed line indicates the number of problems at each stage that exhibit parallelism (more than one plan).

the accuracy of both SPRINT and RFT, albeit with an increase in their average sequential token counts. The accuracy gains are substantial, bringing both models close to the performance of the much larger R1-Distill-32B reasoning model. Notably, SPRINT achieves a higher accuracy of 92.5%, which can be attributed to independent executions within each stage that prevent one result from influencing the others. Despite being fine-tuned on the same trajectories as RFT, reorganized in a plan–execution format, SPRINT requires 440 ($\sim 15\%$) fewer sequential tokens due to parallelized executions. These results demonstrate that SPRINT achieves the same level of reasoning accuracy as conventional distillation used in RFT while substantially reducing the sequential token count.

**Effectiveness of interleaved planning.** The SoT-reasoning baseline underperforms SPRINT in both accuracy and the number of sequential tokens. Since SoT only allows a single round of planning and uses a model without task-specific fine-tuning, it often generates mutually dependent subtasks. When the model executes them independently in parallel, it cannot use the result of one execution to inform another, resulting in redundant computations across subtasks and a total token count that is almost three times higher than SPRINT (see Table 2). Similarly, repeated sampling with self-consistency generates multiple independent responses to the same query, leading to a high total token count. In contrast, SPRINT uses interleaved planning and execution over multiple stages where the plan in each stage is generated based on the results of previous executions, allowing better coordination. Figure 5

| Method | In-domain | | | Out-of-domain | | | |
| | MATH-500 | | | Countdown | | GPQA-Diamond | |
| | Acc↑ | # Seq↓ | # Total↓ | Acc↑ | # Seq↓ | Acc↑ | # Seq↓ |
|---|---|---|---|---|---|---|---|
| Self-consistency | 80.5 | 590 | 11645 | 78.5 | 2845 | 45.4 | 4735 |
| SoT-chat | 47.3 | 256 | 1290 | 80.0 | 2367 | 49.4 | 3526 |
| SoT-reasoning | 90.8 | 3836 | 11538 | 82.4 | 5823 | 48.0 | 7560 |
| RFT | 91.0 | 2880 | 2880 | 84.9 | 4917 | 50.5 | 7103 |
| SPRINT | **92.5** | **2440** | **3622** | **85.9** | **2284** | **51.0** | **6336** |

Table 2: Comparison of pass@1 accuracy and sequential token count across MATH-500, GPQA-Diamond, and Countdown tasks. While SPRINT is only fine-tuned on math reasoning, SPRINT demonstrates strong generalization capabilities on the out-of-domain tasks, Countdown and GPQA-Diamond. SPRINT also reduces sequential token count through parallelized executions without a large increase in total token count.

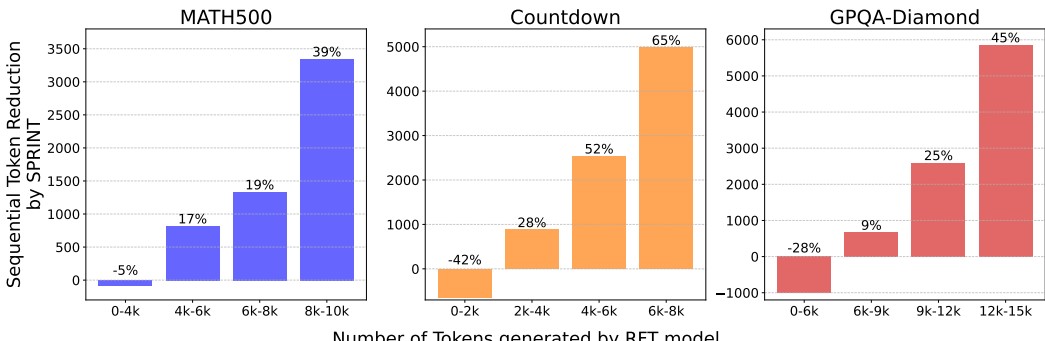

Figure 6: Sequential token reduction achieved by SPRINT. The x-axis shows the number of sequential tokens generated by the RFT baseline model, and the y-axis indicates the average reduction in sequential tokens achieved by SPRINT. As the baseline's sequential requirements increase, SPRINT finds greater opportunities for parallelization, yielding larger sequential token reductions.

illustrates patterns in SPRINT's interleaved planning. As expected, harder problems require more stages before reaching the final answer. Additionally, SPRINT generates more plans in the earlier stages, as the model explores multiple strategies and identifies relevant subtasks, while later stages are more deterministic.

**Reduction in sequential token count.** We further examine the sequential token reduction achieved by SPRINT relative to RFT in Figure 6. For problems with short reasoning trajectories, the additional prompts and plan/execution tags introduce a small overhead, resulting in a 5% increase in sequential tokens. However, as problem difficulty increases and reasoning trajectories become longer, SPRINT consistently reduces the sequential token count relative to the length of the RFT trajectory due to parallel executions. In particular, on problems where RFT requires more than 8,000 tokens on average, SPRINT achieves a 39% reduction in sequential tokens.

**Reduction in runtime.** The savings in sequential tokens translate directly to lower latency. We estimate per-problem runtime by adding the time-to-first-token (TTFT) overhead incurred at the start of each plan/execution to the subsequent decoding time. In practice, decoding dominates; the prefilling (TTFT) cost is comparatively small. Under this estimate, SPRINT outperforms RFT by 9% on MATH-500 (36.92s vs. 40.57s per problem) and by 38% on the subset with longer reasoning chains (74.47s vs. 120.54s). Because runtime scales primarily with the number of decoded tokens, SPRINT's advantage increases with trajectory length, yielding larger absolute and relative latency reductions on harder instances.

**Generalization.** To assess SPRINT's generalization capabilities to out-of-domain tasks, we report performance on Countdown and GPQA-Diamond in Table 2. SPRINT leverages the highly parallelizable nature of the Countdown task to solve problems with much fewer sequential tokens (2284 tokens compared to 4917 tokens by RFT), demonstrating a 53.5% reduction. Notably, these parallelization opportunities are identified despite not being trained on trajectories from this task. Due to the benefits of independent exploration and interleaved planning, SPRINT also beats all baseline methods to achieve an accuracy of 85.9%. Similarly, on the GPQA-Diamond dataset, SPRINT achieves the highest accuracy (51.0%) while reducing sequential token count by 10.8% relative to RFT. Similar to MATH-500, we observe from Figure 6 that SPRINT provides higher efficiency gains on problems with longer reasoning chains.

## 5 Limitations and Future work

**Hardware optimization for realized wall-clock time speed-up.** SPRINT delivers clear efficiency gains, reducing sequential tokens and lowering our end-to-end runtime approximation, but fully realizing these benefits in wall-clock time requires hardware-aware optimizations. Previous works [26, 27, 29] have indicated that sequential token counts are closely correlated with wall-clock latency. However, achieving the ideal latency improvements in practice requires optimized key-value caching mechanisms and high-bandwidth GPU interconnects, especially for long reasoning trajectories

encountered in general tasks. Additionally, executing a large number of parallel tasks simultaneously necessitates a corresponding number of GPUs. Due to limited resources, we were unable to implement the optimal hardware-accelerated decoding for SPRINT. Future work could explore implementing SPRINT within optimized caching frameworks and scalable GPU architectures to fully realize practical wall-clock time efficiency gains offered by parallel decoding strategies.

**Parallelizing tool-use in reasoning models.** In our current work, we primarily treat executions as sequences of tokens that models decode to accomplish tasks. However, from a planning perspective, these executions can alternatively be viewed as black-box modules that receive specific tasks and return corresponding execution results. Several prior works, such as ReAct [23], Self-Ask [34], Swirl [35], and others [36, 37, 38], have introduced mechanisms enabling language models to integrate tool-use into their reasoning loops, iteratively planning, invoking external tools or APIs, and then continuing their reasoning based on the obtained results. Such reasoning-tool interaction trajectories could significantly benefit from parallelization, especially in scenarios where tool invocations dominate the decoding latency. Future work could extend SPRINT's data curation pipeline to accommodate these trajectories, training models to effectively invoke multiple tools or APIs concurrently within their reasoning processes.

**Beyond Supervised Training.** Through supervised fine-tuning (SFT) on curated data, our model learned how to define parallelizable plans, effectively reducing sequential token generation. However, the achievable parallelism is inherently limited by the quality of training data. Future work could explore latency-aware reinforcement learning (RL), using reward signals based on inference efficiency, allowing models to autonomously discover strategies that further enhance parallel reasoning beyond the constraints of demonstration data.

# 6   Conclusion

In this work, we presented SPRINT, a framework for post-training reasoning language models that reorganizes their reasoning trajectories into a series of plans and parallelized executions. Additionally, SPRINT introduces an inference mechanism that leverages the trained reasoning model to identify independent subtasks and execute them in parallel. This approach significantly reduces the number of sequential tokens while achieving comparable state-of-the-art performance to the reasoning fine-tuned (RFT) model. Notably, on problems requiring extensive reasoning trajectories, SPRINT uncovers even greater parallelization potential, achieving sequential token reductions of 39%. Furthermore, we evaluated our model's generalization on multiple out-of-domain tasks and consistently found that SPRINT generates substantially fewer sequential tokens while maintaining performance on par with RFT. These results suggest that the SPRINT training unlocks parallelized reasoning capabilities in the model across diverse domains with longer reasoning trajectories.

# 7   Acknowledgment

This work was supported in part by the Air Force Office of Scientific Research (AFOSR) under Grant FA9550-23-1-0251 and in part by the Office of Naval Research under Grant N00014-24-1-2164. We also thank Yuhao Ge at the University of Illinois Urbana-Champaign for his guidance on the model training process and computing requirements.

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

# A   Implementation Details

## A.1   Inference

**Inference from a sequential reasoning model.**   To generate responses from sequential reasoning models, such as DeepSeek-R1-Distill-7B and the RFT model, we use the prompt provided below. The same prompt was used for fine-tuning DeepSeek-R1-Distill-7B to derive the RFT model. During inference, the question is appended to the prompt and the model is called in the completions format. Following guidelines suggested by DeepSeek [3], we set the generation temperature to 0.6 to mitigate repetitive outputs. Additionally, we enforce a maximum token limit of 36,000 per response, truncating any outputs exceeding this threshold.

---

**Sequential Reasoning Prompt**

Your role as an assistant involves thoroughly exploring questions through a systematic long thinking process before providing the final precise and accurate solutions. This requires engaging in a comprehensive cycle of analysis, summarizing, exploration, reassessment, reflection, backtracking, and iteration to develop well-considered thinking process.

Please structure your response into two main sections: **Thought** and **Solution**.

- In the Thought section, detail your reasoning process using the specified format: `<think>` {thought with steps separated with "\n \n"} `</think>` Each step should include detailed considerations such as analyzing questions, summarizing relevant findings, brainstorming new ideas, verifying the accuracy of the current steps, refining any errors, and revisiting previous steps.
- In the Solution section, based on various attempts, explorations, and reflections from the Thought section, systematically present the final solution that you deem correct. The solution should remain a logical, accurate, concise expression style and detail necessary step needed to reach the conclusion.

Now, try to solve the following question through the above guidelines. Return your final response within `\boxed{}`.

---

**SPRINT Inference.**   During inference, we use the following prompt to guide the generation of plans and executions from the SPRINT model. Although the model is fine-tuned to produce an entire trajectory—including all plans and executions—in a single generation, we manage model invocations and output token handling to alternate between planner and executor roles effectively.

To restrict the model's outputs to either a single plan or execution per invocation, we employ specific stop tokens. Generation is terminated once the model produces any of the following strings, indicating the completion of a plan or execution segment: {`</Execution_`, `</Plan_`, `</Final_answer>`, `</execution_`}.

When generating a plan for stage $i$, we feed the prompt along with the input query and the cumulative context, which includes all preceding plans and executions, to the SPRINT model. Conversely, to generate the execution corresponding to a particular `prompt_i.j`, we provide the model with the prompt, the input query, all previously generated plans and executions up to stage $i - 1$, and the text from plan $i$ until the end of `prompt_i.j`. This structured context management allows us to reuse the same prompt for both planning and execution tasks seamlessly.

The model is permitted a maximum of 12 stages to produce a final answer. To enforce this constraint, we append "`<Final_answer>\n`" at the end of the prompt when invoking the model at the 12th stage. Generated responses for each plan or execution are limited to 8,000 tokens, with any excess tokens truncated accordingly. This prompt is identical to that used during model fine-tuning.

## A.2 Dataset Curation

**Step Extraction.** For step extraction, we use the GPT-4o model with the temperature set to 0. The following prompt is used to extract steps from a reasoning trajectory generated by DeepSeek R1. In the prompt, we use the term "Component" to refer to the extracted steps to prevent the model from confusing it with the traditional use of the term "step" in a math solution which could be a single operation as opposed to a logical part of the solution. Components, as defined here, may involve tasks such as identifying subsequent actions, validating previous results, proposing alternative methods, or comparing solutions derived through different strategies. For each component, the model starts by thinking out loud about what needs to be done and then carries out the identified task. We refer to the first part as the plan and the second as the execution and extract them separately using this prompt.

The reasoning trajectory passed to the model as input is formatted by labeling each line/sentence with a unique line number and the model provides the range of line numbers for each plan and execution within a component. This minimizes the number of output tokens that have to be generated by the model, consequently reducing costs. The line numbers are later parsed from the response to infer the block of text that is relevant to each plan or execution.

- **Execution**: Here, the model follows the plan and executes it.
- **Commenting**: Here, the model comments on the execution results with phrases such as "Yes, that seems right", "Both methods lead to the same answer, etc.

Note that the verification of an execution should be considered as a separate component and not as the commenting phase of the same component.

I am building a new AI system to solve such math problems. This system will consist of two separate AI models – a planner and an executor.

- **Planner**: The planner will receive all the components of the solution completed so far and will need to think aloud and generate a plan for the next component. Then, it needs to provide a prompt to the executor model to execute a specific task.
- **Executor**: The executor will receive all the components of the solution completed so far, the plan for the next component generated by the planner, and the prompt generated by the planner. It will need to execute the specified task.

To train these two AI models, I must generate training data by breaking down the solution provided below into individual components. For each component, clearly provide the following details:

> **Required Response Format:**
> ### Component X (Line Number Range)
>
> - **Description**: Brief explanation of what this component achieves.
> - **Plan**: Lines (minimal number of lines to describe the plan clearly).
> - **Prompt**: A precise, actionable instruction for the executor based explicitly on the above plan.
> - **Execution**: Lines (specific line numbers performing the planned task).
> - **Comment**: Lines (reflective comments or **Lines not found** if missing).

**Important Notes:**

- The planning phase should only include a minimal number of lines required to specify what needs to be done. The remaining lines from the component where the model carries out the plan should be included in the execution phase.
- There MUST be NO overlap between the line numbers of different components.
- There MUST be NO overlap between the line numbers of the planning and execution phases of the same component.
- All the lines in the solution should be covered by the components.
- Use the line number mentioned at the start and end of each line to identify the line when specifying the line number range.
- The prompt to the executor model must be a very specific instruction that the executor can follow to complete the required task. The executor must not perform more tasks than required. The prompt can refer to the plan for that component by saying "the above plan".
- If the model does not comment on the execution results within a component, the corresponding bullet point can be written as **Comment**: Lines not found

**DAG Creation.**    For DAG creation, we use the GPT-4o-mini model with the temperature set to
0. The following prompt is used to infer the DAG in the form of a parent dictionary, where each
key refers to a step extracted above and the corresponding values refer to the steps on which the key
depends.



**DAG Creation Prompt**

Given below is a well-thought out solution to a math problem generated by an AI system.
The system consists of a **planner** and an **executor**. The planner model thinks out loud and
plans the next component of the problem solution. Then, it provides a prompt along with the
plan to an executor model. The executor then follows the instructions in the prompt and uses
context from the plan to carry out the given task.
The solution consists of multiple **components**, each containing the following:

- **Description**: A brief description of what the component does.

- **Plan**: The plan generated by the planner.

- **Prompt**: Instructions generated by the planner enclosed within `<prompt>` tags.

- **Execution**: Output provided by the executor.

Though the executions are run sequentially in this solution, some of the executions may be
parallelized to improve speed. Identify and explain which components can run in parallel and
determine the best way to parallelize them to maximize speed. Note that parallel runs should
not have co-dependency.

The parallelization schedule can be represented as a **directed acyclic graph (DAG)** where
the nodes are the component numbers. You need to represent the DAG as a parent dictionary
where each node is a key and its value is a list of nodes that point to it, i.e., the nodes that
must be executed immediately before it. For a key node, do not include any nodes in its value
that can be run in parallel with it.

**Format of parent dictionary:**
Let us consider a simple example. Suppose that the following constraints hold:

- Component 1 needs to be run before any other component

- Components 2, 3, 4 can be run in parallel after 1

- Component 5 which depends on the results of 2 and 3 can be run after 2 and 3

- Component 6 which depends on the results of 4 and 5 can be run after 4 and 5

The parent dictionary for this example *MUST* be represented as a python dictionary as
follows:

```
parent_dictionary = {
    1: [],
    2: [1],
    3: [1],
    4: [1],
    5: [2, 3],
    6: [4, 5]
}
```



Using the resulting DAG, we can reorganize the components into interleaved plans and executions to
obtain a parallelizable reasoning trajectory. A simple strategy involves assigning components at the
same DAG depth to the same planning-execution stage. However, further optimization can reduce the
total number of stages required to reach the final answer.

**Packing.**    The objective of packing is to optimally assign stage numbers to each component. To
achieve this, we apply the following greedy heuristics:

- If a component's execution consists of fewer than three lines, it is merged directly with its corresponding plan. This approach reduces overhead from additional prompt writing and executor invocation. Through fine-tuning on trajectories with merged short executions, the planner learns to carry out short or trivial executions on its own.

- If a component $C$ depends on a plan-only component $P$, then $C$'s plan is independent of the execution results from $P$'s stage. When all of $C$'s parent components satisfy this condition, $C$ is merged into the same planning stage as $P$ by combining their respective plans.

As a result, we obtain optimal stage numbers for each component which can then be used for generating the fine-tuning trajectory.

### A.3 Fine-tuning

We conducted supervised fine-tuning (SFT) of our models by training on the reasoning trajectories. Initially, we experimented with more efficient fine-tuning techniques such as LoRA[39] and qLoRA [40]. However, since LoRA did not adequately enable the models to adhere to the desired response format, we proceed with full fine-tuning instead.

Fine-tuning was primarily executed on a single machine with eight NVIDIA A100 GPUs with 40 GB memory per GPU. We use the `ms-swift` framework [41], a fine-tuning toolkit provided by the Modelscope community.

Each model is fine-tuned for 5 epochs. Due to the long-context required for reasoning traces and the memory constraints, we use a batch size of 1 during the training. We use `bfloat16` precision, an initial learning rate of $1 \times 10^{-5}$, and a weight decay factor of $1 \times 10^{-4}$. The learning rate scheduling consists of a linear warm-up phase during the first 5% of training steps, subsequently followed by linear decay to zero over the remaining training iterations. Model evaluation is conducted every 100 steps, and the best-performing model based on evaluation loss is retained.

To optimize memory usage during training, we integrate several efficiency strategies, notably the DeepSpeed ZeRO Redundancy Optimizer [42, 43] and 4-bit quantization. DeepSpeed's ZeRO optimizer offers a set of memory-partitioning strategies that trade off memory savings against communication overhead. In many workloads, ZeRO Stage 1 or 2 strikes the best balance between memory efficiency and communication cost; however, since we need to train on long sequences, our per-GPU memory demands exceed what those stages can support. Therefore, we adopted ZeRO Stage 3 to train with extended context lengths without OOM errors.

### A.4 Evaluation

For model evaluation, we leverage vLLM [44] to serve our models. Specifically, each 7B-scale model (SPRINT, RFT, and DeepSeek-R1-Distill-7B) is deployed on a single NVIDIA A100 GPU with 40 GB of memory.

To enhance evaluation accuracy, we instruct the models to encapsulate their final answers within `\boxed{}`. For evaluations on the MATH-500 and Countdown tasks, we leverage the Math-Verify library alongside SymPy for equivalence checking, ensuring robustness against mathematically equivalent but differently expressed solutions. In the GPQA task, accuracy is determined by comparing explicitly generated option labels (e.g., A, B, C, D) directly with the corresponding ground-truth options.

Despite providing explicit formatting instructions, we occasional observe deviations by the models from the specified output format. For instance, during the Countdown task evaluation, the models occasionally produce outputs in unexpected formats (e.g., `\times`, ×), complicating the parsing procedure. To mitigate these issues, we incorporate the GPT-4o-mini model as an auxiliary evaluator to judge the generated answer against the ground truth.

### A.5 Baselines

Below we present the implementation details for the baselines used in our study:

- **SoT (Skeleton of Thought):** For SoT-chat, we use the Qwen2.5-7B-Instruct model with the temperature set to 0.1. For SoT-reasoning, we use the DeepSeek-R1-Distill-Qwen-7B

model with the temperature set to 0.6. We use the same prompts as reported in the original paper [25]. After the full solution is generated, we send it to GPT-4o-mini to extract the final answer for evaluation.

- **RFT (Reasoning Fine-tuned Model):** This baseline was created by fine-tuning the DeepSeek model on sequential R1 trajectories. During inference, we set the temperature to 0.6 and employed the sequential reasoning prompt provided in Appendix A.1.

- **Parallel Sampling + Self-consistency:** The Qwen2.5-7B-Instruct model was used with a temperature of 0.3, generating 20 independent samples per problem. The majority answer is considered as the predicted answer for the evaluation of this baseline.

# B   Sample Demonstrations

## B.1   Demonstration of the Data Curation Pipeline

To illustrate our data curation pipeline, we examine a representative example from the MATH training dataset. Consider the following question:

**Question:** The vertex of the parabola described by the equation $y = -3x^2 - 30x - 81$ is $(m, n)$. What is $n$?

Next, we provide the response generated by DeepSeek-R1 for this question with each logical component in a separate box. Note that the original response does not provide any separation among its logical components and the full text is generated as a single long trajectory.

---

### Outline of DeepSeek-R1 response

Okay, so I need to find the vertex of the parabola given by the equation $y = -3x^2 - 30x - 81$, and specifically determine the $n$-coordinate of the vertex, which is $(m, n)$. Hmm, let's start by recalling how to find the vertex of a quadratic equation. I remember that quadratic equations are in the form $y = ax^2 + bx + c$, and their graphs are parabolas. The vertex of a parabola can be found using the formula for the $x$-coordinate, which is $-b/(2a)$. Then once we have the $x$-coordinate, we can plug that back into the original equation to find the corresponding $y$-coordinate, which would be $n$ in this case.

So let's identify $a$, $b$, and $c$ from the given equation. Comparing $y = -3x^2 - 30x - 81$ with the standard form $y = ax^2 + bx + c$, it looks like $a = -3$, $b = -30$, and $c = -81$.

---

Wait, is that right? Let me check again. The coefficient of $x^2$ is $-3$, so $a = -3$. The coefficient of $x$ is $-30$, so $b = -30$. The constant term is $-81$, so $c = -81$. Okay, that seems correct.

---

Now, using the formula for the $x$-coordinate of the vertex: $m = -b/(2a)$. Plugging in the values of $a$ and $b$, we get $m = -(-30)/(2 * (-3))$. Let's compute that step by step. The numerator is $-(-30)$, which is positive 30. The denominator is 2 times $a$, which is $2 * (-3) = -6$. So $m = 30/(-6) = -5$. Therefore, the $x$-coordinate of the vertex is $-5$. That seems straightforward.

---

Now, to find the $y$-coordinate $n$, I need to substitute $x = -5$ into the original equation. Let's do that. The original equation is $y = -3x^2 - 30x - 81$. Plugging in $x = -5$:

$y = -3 * (-5)^2 - 30 * (-5) - 81$. Let's compute each term step by step. First, $(-5)^2$ is 25, so $-3 * 25$ is $-75$. Then, $-30 * (-5)$ is positive 150. And the last term is $-81$. So adding them up: $-75 + 150 - 81$. Let's calculate this.

First, $-75 + 150 = 75$. Then, $75 - 81 = -6$. So $n = -6$. Therefore, the vertex is at $(-5, -6)$, so $n$ is $-6$.

---

Wait, let me double-check my calculations to make sure I didn't make any arithmetic errors. Let's redo the substitution.

Calculating $y$ when $x = -5$:

First term: $-3 * (-5)^2$. The exponent first: $(-5)^2 = 25$. Then multiply by $-3$: $-3 * 25 = -75$. Second term: $-30 * (-5)$. Multiply those: $-30$ times $-5$ is $150$. Third term is $-81$. So adding them: $-75 + 150 = 75$; $75 - 81 = -6$. Yep, that's correct.

Alternatively, maybe I can use another method to confirm. Sometimes, completing the square can be a good way to find the vertex. Let me try that. The given equation is $y = -3x^2 - 30x - 81$. To complete the square, first factor out the coefficient of $x^2$ from the first two terms. So:

$y = -3(x^2 + 10x) - 81$. Wait, let's check: $-3x^2 - 30x$ becomes $-3(x^2 + 10x)$. Yes, that's correct because $-3$ times $x^2$ is $-3x^2$ and $-3$ times $10x$ is $-30x$. Then, we have the remaining constant term as $-81$.

Now, to complete the square inside the parentheses, we need to take half of the coefficient of $x$, which is 10, so half of 10 is 5, and then square it: $5^2 = 25$. So we add and subtract 25 inside the parentheses to maintain equality. However, since we are adding 25 inside the parentheses that's multiplied by $-3$, we have to balance that by actually adding and subtracting $-3 * 25$ to the entire equation. Wait, maybe I need to write it out more carefully.

So:

$y = -3(x^2 + 10x + 25 - 25) - 81$

Wait, that is equivalent to:
$y = -3[(x^2 + 10x + 25) - 25] - 81$

Which simplifies to:
$y = -3(x + 5)^2 + 75 - 81$

Because $(x^2 + 10x + 25)$ is $(x + 5)^2$, and then we have $-3$ times $-25$ which is $+75$. Then, subtract 81. So:
$y = -3(x + 5)^2 + 75 - 81$

Then, combining the constants: $75 - 81 = -6$. So the equation becomes:
$y = -3(x + 5)^2 - 6$

Which is in vertex form: $y = a(x - h)^2 + k$, where the vertex is $(h, k)$. Wait, in this case, it's $(x + 5)$ which is the same as $(x - (-5))$, so $h$ is $-5$ and $k$ is $-6$. Therefore, the vertex is at $(-5, -6)$, so $n$ is $-6$. That confirms the previous result.

Alternatively, maybe I can use calculus, taking the derivative and setting it to zero, but since this is a simple quadratic, that might be overkill. But for thoroughness, let's try it. The derivative of $y$ with respect to $x$ is $dy/dx = 2 * (-3)x + (-30) = -6x - 30$. Setting this equal to zero for critical points: $-6x - 30 = 0$. Solving for $x$: $-6x = 30 \Rightarrow x = -5$. Then substitute $x = -5$ into original equation, as before, gives $y = -6$. So again, same result.

> Therefore, all three methods (vertex formula, completing the square, calculus) lead to the same answer: $n = -6$.

> Is there any chance I made a mistake in the signs? Let me check again. The original equation is $y = -3x^2 - 30x - 81$. So $a = -3$, $b = -30$. Then the $x$-coordinate of vertex is $-b/(2a) = -(-30)/(2*(-3)) = 30/-6 = -5$. That's correct. Then substituting $x = -5$ into the equation: $y = -3*(-5)^2 - 30*(-5) - 81$. Compute each term:
>
> First term: $-3*(25) = -75$
>
> Second term: $-30*(-5) = +150$
>
> Third term: $-81$
>
> Total: $-75 + 150 = 75$; $75 - 81 = -6$. Correct.
>
> Completing the square also confirmed the same. Calculus as well. So multiple methods confirm that $n = -6$.
>
> Therefore, I'm confident that the answer is $n = -6$.

Next, we demonstrate the output of step extraction. As explained in A.2, the output from step extraction contains line numbers corresponding to each plan and execution within a component. In the sample below, we replace line numbers with the corresponding block of text from the reasoning trajectory for ease of understanding. For the sake of brevity, we only show the full details for a few components.

---

**Formatted output of Step Extraction**

### Component 1
**Description:** Planning how to find the vertex of the parabola.
**Plan:** Okay, so I need to find the vertex ... Then once we have the $x$-coordinate, we can plug that back into the original equation to find the corresponding $y$-coordinate, which would be $n$ in this case.
**Prompt:** Identify the values of $a$, $b$, and $c$ from the given quadratic equation.
**Execution:** So let's identify $a$, $b$, and $c$ ... it looks like $a = -3$, $b = -30$, and $c = -81$.
**Comment:** No lines found

### Component 2
**Description:** Verifying the identified values of $a$, $b$, and $c$.

...

### Component 3
**Description:** Calculating the $x$-coordinate of the vertex using the vertex formula.

...

### Component 4
**Description:** Calculating the $y$-coordinate of the vertex by substituting the $x$-coordinate.
...

```
### Component 5
Description: Verifying the calculation of the y-coordinate.
...

### Component 6
Description: Using the method of completing the square to find the vertex.
Plan: Alternatively, maybe I can use another method to confirm. Sometimes, completing the
square can be a good way to find the vertex. Let me try that.
Prompt: Use the method of completing the square on the given equation to find the vertex.
Execution: The given equation is y = -3x² - 30x - 81. To complete the square, first factor
out the coefficient of x² ... So the equation becomes: y = -3(x + 5)² - 6 which is in vertex
form: y = a(x - h)² + k, where the vertex is (h, k) ... Therefore, the vertex is at (-5, -6),
so n is -6.
Comment: That confirms the previous result.
...

### Component 7
Description: Using calculus to find the vertex by taking the derivative and setting it to zero.
...

### Component 8
Description: Comparing results from different methods.
...

### Component 9
Description: Final verification of the solution and confirming results.
...
```

In Figure 7, we demonstrate the dependencies that are inferred from running DAG creation over the steps (a.k.a components) extracted above. The model has identified that Components 2, 3, 6, and 7 can run in parallel. This is because once the values of $a$, $b$, and $c$ are identified in Component 1, the verification of that calculation (Component 2) and the computation of the $x$-coordinate (Component 3) are mutually independent. Besides, alternative approaches such as completing the square (Component 6) and using calculus (Component 7) are not dependent on components other than Component 1. Component 4 depends on Component 3 as the computation of the $y$-coordinate depends on the value of the $x$-coordinate identified in Component 3. Comparing the results of the three approaches (Component 8) depends on the results from Components 4, 6, and 7 while the final verification (Component 9) depends on Component 8. Hence, the DAG effectively models the dependencies among the components and identifies parallelization opportunities.

As explained in Appendix A.2, for components with short executions, we merge the executions with their plans. Hence, for the above problem, the executions of Components 2, 3, and 8 are merged with their plans. The trajectory containing plans and executions generated for the above problem following this modification is given below. Note that the plans are ordered based on the dependencies modeled in the DAG.

The sequential token count for the original response from R1 is measured as the total number of tokens in the response which is 1645. The trajectory sample below has a smaller sequential token count of 1445 due to the parallelized executions.

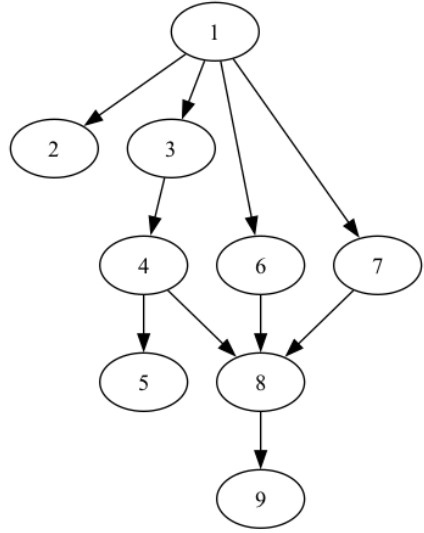

Figure 7: Dependencies inferred from running DAG creation over the steps extracted from the example reasoning trajectory in Appendix B.1

---

**Trajectory sample for fine-tuning**

```
<Plan_1>
```

Okay, so I need to find the vertex ... it looks like $a = -3$, $b = -30$, and $c = -81$.
...
Wait, is that right? Let me check again ... The constant term is $-81$, so $c = -81$.
...
Now, using the formula and plugging in the values of $a$ and $b$... Let's compute that step by step.
...
`<prompt_1.1>` **Calculate the x-coordinate of the vertex** `</prompt_1.1>`

Alternatively, maybe I can use another method to confirm. Sometimes, completing the square can be a good way to find the vertex.
`<prompt_1.2>` **Use the method of completing the square on the given equation to find the vertex.** `</prompt_1.2>`

Alternatively, maybe I can use calculus, taking the derivative ... let's try it.
`<prompt_1.3>` **Use calculus to find the $x$-coordinate** `</prompt_1.3>`

```
</Plan_1>

<Execution_1>
```

```
<execution_1.1>
```
The numerator is $-(-30)$... So $m = 30/(-6) = -5$. Therefore, the $x$-coordinate of the vertex is $-5$.
```
</execution_1.1>
```

```
<execution_1.2>
```
The given equation is $y = -3x^2 - 30x - 81$. To complete the square, first factor out the coefficient of $x^2$ ... So the equation becomes: $y = -3(x + 5)^2 - 6$ which is in vertex form: $y = a(x - h)^2 + k$, where the vertex is $(h, k)$ ... Therefore, the vertex is at $(-5, -6)$, so $n$ is $-6$.
```
</execution_1.2>
```

```
<execution_1.3>
```
The derivative of $y$ with respect to $x$ is $dy/dx = 2 * (-3)x + (-30) = -6x - 30$. Setting this equal to zero for critical points ... as before, gives $y = -6$.
```
</execution_1.3>
```

```
</Execution_1>
```

```
<Plan_2>
```

Now, to find the y-coordinate $n$ ... , I need to substitute $x = -5$...
```
<prompt_2.1>
``` **Substitute x = -5 to find the y-coordinate of the vertex.**
```
</prompt_2.1>
```
```
</Plan_2>
```

```
<Execution_2>
```
```
<execution_2.1>
```
Plugging in x = -5:

...
So $n = -6$.
```
</execution_2.1>
```
```
</Execution_2>
```

```
<Plan_3>
```
Based on execution_2.1:
Wait, let me double-check my calculations ...
```
<prompt_3.1>
``` **Redo the substitution of $x = -5$ into the original equation to verify.** ```
</prompt_3.1>
```

Based on execution_2.1, execution_1.2, execution_1.3:
Therefore, all three methods (vertex formula, completing the square, calculus) lead to the same answer: $n = -6$.

Let me check again.
```
<prompt_3.2>
``` **Recheck the calculations and confirm the results** ```
</prompt_3.2>
```

```
</Plan_3>
```

```
<Execution_3>
```

```
<execution_3.1>
```
Calculating $y$ when $x = -5$: First term: $-3 * (-5)^2$ ... So adding them: $n = -75 + 150 = 75; 75 - 81 = -6$.
```
</execution_3.1>
```

```
<execution_3.2>
```
The original equation is ... So multiple methods confirm that $n = -6$.
```
</execution_3.2>
```

```
</Execution_3>
```

```
<Final_answer>
```

Therefore, the value of $n$ is \boxed{-6}.

```
</Final_answer>
```

## B.2 Demonstration of SPRINT execution versus sequential reasoning from RFT

Given below is an example of how SPRINT responds to a question from the MATH-500 benchmark at inference time. The executions presented side-by-side are carried out in parallel using the inference strategy shown in Figure 3.

**Question:** The binary number $10101001110_2$ is equal to what number in base eight?

---

**Planning and parallel execution trajectory generated by SPRINT**

```
<Plan_1>
```

Okay, so I need to convert the binary number $10101001110_2$ to base eight. So, the plan is probably to group the binary digits into sets of three starting from the right and then convert each group to an octal digit.

Wait, but first, let me confirm: the binary number given is 10101001110. Let me count the digits.

Let's see: 10101001110. That's 11 binary digits. Since 8 is $2^3$, each octal digit is three bits, so 11 bits would need to be grouped into 4 octal digits (since 4*3=12, which is more than 11) but we need to add a leading zero to make it 12 bits? Wait, no, actually, when converting from binary to octal, you start from the right and split into groups of three. Right, so even if the total number of bits isn't a multiple of three, you can add leading zeros to make it a multiple of three. Let me check.

`<prompt_1.1>` **Verify the method of converting binary to octal by considering the binary number** 10101001110**.** `</prompt_1.1>`

But let me think again. Let me confirm with another approach. Alternatively, I could convert the binary number to decimal first and then to octal. Maybe that's a good way to cross-verify.

`<prompt_1.2>` **Consider converting the binary number to decimal first and then to octal as an alternative method.** `</prompt_1.2>`

```
</Plan_1>

<Execution_1>
```

`<execution_1.1>`
The original number is 10101001110. Let's write them out: positions from right to left (since we start grouping from the right). Wait, actually, when grouping, we start from the right, so the leftmost digit might need to be padded with zeros to make the total number of digits a multiple of three. So the binary number has 11 digits. 11 divided by 3 is 3 with a remainder of 2. So, we need to add one leading zero to make it 12 digits. Wait, 3*4=12, so yes, adding one zero on the left. So the original binary is 10101001110. Adding a leading zero gives 010101001110.
`</execution_1.1>`

`<execution_1.2>`
Let's see. The binary number is 10101001110. Let me write down the positions from right to left, starting at 0. Wait, actually, in computers, the rightmost bit is the least significant bit, which is position 0. So let's list the bits:
Position: 10 9 8 7 6 5 4 3 2 1 0 Bit: 1 0 1 0 1 0 0 1 1 1 0 ...
So: Bit 10: 1 ($2^{10}$)
Bit 9: 0 ($2^9$)
Bit 8: 1 ($2^8$)
...
So the binary number is: $2^{10}+2^8+...2^1$
... ... Adding them all up: 1024 + 256 = 1280; 1280 + 64 = 1344; 1344 + 8 = 1352; 1352 + 4 = 1356; 1356 + 2 = 1358. So the decimal equivalent is 1358. `</execution_1.2>`

```
    </Execution_1>

    <Plan_2>
        Based on execution_1.1:
        Now, splitting into groups of three starting from the right:
        <prompt_2.1> Split the 12-digit binary number into groups of three starting
        from the right and convert each group to octal. </prompt_2.1>

        Based on execution_1.2:
        Now, converting 1358 to octal. Let's divide by 8:
        <prompt_2.2> Convert the decimal number 1358 to octal using division by 8.
        </prompt_2.2>

    </Plan_2>

    <Execution_2>
```

<execution_2.1>
0 1 0 1 0 0 0 1 1 1 0
Wait, no, wait. Wait, the binary number after adding a leading zero is 010101001110. Now, group into 4 sets of three: Group 1 (leftmost): 010, Group 2: 101, Group 3: 001, Group 4: 110. Then convert each of these to octal digits. Wait, octal digits go from 0 to 7, so each group of three bits represents 0-7. Let me recall the binary to octal mapping:
$000 = 0, 001 = 1, 010 = 2, ...$ So:
Group 1: $010 \rightarrow 2$
Group 2: $101 \rightarrow 5$
Group 3: $001 \rightarrow 1$
Group 4: $110 \rightarrow 6$
So putting them together: $2\ 5\ 1\ 6 \rightarrow 2516_8$.
</execution_2.1>

<execution_2.2>
$1358 \div 8 = 169$ with remainder 6 (since 169*8 = 1352, 1358 - 1352 = 6)
$169 \div 8 = 21$ with remainder 1 (21*8 = 168, 169 - 168 = 1)
$21 \div 8 = 2$ with remainder 5 (2*8 = 16, 21 - 16 = 5)
$2 \div 8 = 0$ with remainder 2
So the remainders, from last to first, are 2, 5, 1, 6. Therefore, 1358 in octal is $2516_8$.
</execution_2.2>

```
    </Execution_2>

    <Plan_3>

        Based on execution_2.1:
        Wait, but let me check again. Let's verify the grouping.
        <prompt_3.1> Verify the grouping of the binary digits and the conversion to
        octal. </prompt_3.1>

        Based on execution_2.2:
```
Therefore, the binary number $10101001110_2$ is equal to $2516_8$. So both methods give the same result, which is reassuring.
```
        <prompt_3.2> Reflect on the solution and confirm its correctness by comparing
        both conversion methods. </prompt_3.2>

    </Plan_3>

    <Execution_3>
```

<execution_3.1>
Original binary with leading zero: 010 101 001 110.
Group 1: $010 \rightarrow 2$
Group 2: $101 \rightarrow 5$
Group 3: $001 \rightarrow 1$
Group 4: $110 \rightarrow 6$
So octal number is 2 5 1 6 $\rightarrow$ 2516.
</execution_3.1>

<execution_3.2>
I think this confirms that the answer is correct. But let me just go through the steps once more to be thorough. Original binary: 10101001110 Convert to groups of three starting from the right. Since it's 11 digits, add a leading zero to make it 12 digits: 010101001110.
Group into four parts: 010 101 001 110. Convert each:
$010 = 2, 101 = 5, 001 = 1, 110 = 6$
Combine: 2516. Yep.
Alternatively, converting to decimal via binary to decimal conversion: $10101001110_2 = 1 * 2^10 + 0 * 2^9 + 1 * 2^8 + 0 * 2^7 + ... + 0 * 2^0 = 1024 + 256 + 64 + 8 + 4 + 2 = 1358$. Then 1358 divided by 8 gives 169*8=1352, remainder 6; 169 divided by 8 is 21*8=168,
... So 2 5 1 6. So 2516. All checks out. Therefore, the final answer is 2516 in octal. I think that's solid. No mistakes detected in the process.
</execution_3.2>

</Execution_3>

<Final_answer>

This step-by-step conversion shows that the binary number $10101001110_2$ is equal to $\boxed{2516_8}$ in base eight.

</Final_answer>

The sequential token count of the above response from SPRINT is 1463. For the same question, we present the sequential trajectory generated by the RFT model below. The sequential token count of this response is 1989 since all tokens are generated sequentially. This highlights the benefit of parallelizing executions using SPRINT.

**Sequential reasoning trajectory generated by the RFT model**

Okay, so I need to convert the binary number $10101001110_2$ to base eight. Hmm, I remember that converting between binary and octal is done by grouping the binary digits into sets of three, starting from the right. Each group of three bits corresponds to a single octal digit. Let me think through the steps carefully.

First, let me write down the binary number: 1 0 1 0 1 0 0 1 1 1 0. Wait, let me count the digits. The original number is 10101001110, which has 11 digits. Since octal groups are in threes, I need to split this into groups of three. But 11 isn't divisible by three. So, I should add leading zeros to make the total number of digits a multiple of three. Right, so 11 divided by 3 is 3 with a remainder of 2. That means I need to add one leading zero to make it 12 digits. So the new number becomes $010101001110_2$.

Now, let me split this into groups of three starting from the right. Let me write it out:

Group 1: 010, Group 2: 101, Group 3: 001, Group 4: 110

Wait, no. Wait, the leading zeros are added at the beginning, so the original number was 10101001110. Adding a leading zero makes it 010101001110. Now, splitting into groups of

three from the right:

Starting from the end, the last three digits are 110, then moving left, the next three are 001, then 101, then 010. So the groups are 010, 101, 001, 110. Now, each of these binary groups needs to be converted to their octal equivalents.

Let me convert each group:

$010_2$ is 2 in octal because $0 \times 4 + 1 \times 2 + 0 \times 1 = 2$.

$101_2$ is 5 in octal because $1 \times 4 + 0 \times 2 + 1 \times 1 = 5$.
...

So putting those together in order from left to right, the octal digits are 2, 5, 1, 6. Therefore, the octal number should be $2516_8$. Wait, but let me verify this again to make sure I didn't make a mistake in grouping or conversion.

Wait, when grouping from the right, the first group (rightmost) is 110, then moving left, the next is 001, then 101, then 010. So the groups are 010, 101, 001, 110. Converting each to octal:

$010_2 = 2, 101_2 = 5, 001_2 = 1, 110_2 = 6$. So the octal number is 2 5 1 6, which is $2516_8$. Let me check if this is correct by converting the original binary number to decimal and then converting that decimal number to octal to see if it matches.

First, converting $10101001110_2$ to decimal. Let's write out the binary number with each digit's place value:

From right to left, the positions are 0 to 10. The digits are:

Position 10: 1, Position 9: 0, Position 8: 1, ..., Position 0: 0

Calculating the decimal value:

$1 \times (2^{10}) + 0 \times (2^9) + 1 \times (2^8) + 0 \times (2^7) + 1 \times (2^6) + 0 \times (2^5) + 0 \times (2^4) + 1 \times (2^3) + 1 \times (2^2) + 1 \times (2^1) + 0 \times (2^0)$

Calculating each term:

$2^{10} = 1024, 2^8 = 256, 2^6 = 64, 2^3 = 8, 2^2 = 4, 2^1 = 2$

Adding them up: $1024 + 256 = 1280; 1280 + 64 = 1344; 1344 + 8 = 1352; 1352 + 4 = 1356; 1356 + 2 = 1358$.

So the decimal value is 1358. Now, converting 1358 to octal. Let's divide 1358 by 8 repeatedly.

$1358 \div 8 = 169$ with a remainder of 6.
$169 \div 8 = 21$ with a remainder of 1.
$21 \div 8 = 2$ with a remainder of 5.
$2 \div 8 = 0$ with a remainder of 2.

Writing the remainders from last to first: 2 5 1 6. So $2516_8$. Which matches the previous result. Therefore, the conversion is correct.

But wait, let me check the grouping again. The original binary number was 10101001110. When adding a leading zero, it becomes 010101001110. Splitting into groups of three from

the right:

Starting from the right:
111 001 101 010. Wait, that's different. Wait, perhaps I made a mistake in grouping.

Wait, the original binary number is 10101001110. Let me write it as 10 101 001 110. Wait, that's not correct. Wait, the original number is 11 digits. So when adding a leading zero, it becomes 12 digits: 0 1 0 1 0 1 0 0 1 1 1 0. Then grouping from the right:

Group1: 0 1 0 (positions $11, 10, 9$), Group2: 1 0 1 (positions $8, 7, 6$), Group3: 0 0 1 (positions $5, 4, 3$), Group4: 1 1 0 (positions $2, 1, 0$)

Wait, but when grouping from the right, the rightmost group is the last three digits. So the original number is 010101001110. The rightmost three digits are 110, then moving left, next three are 001, then 101, then 010. So the groups are 010, 101, 001, 110. Which is the same as before. So converting each gives $2, 5, 1, 6$. So $2516_8$. Which is correct. So the answer is $2516_8$.

But wait, when I grouped the original number as 10101001110, which is 11 digits, adding a leading zero makes it 010101001110, which is 12 digits. So grouping into three from the right: 010, 101, 001, 110. So that's correct.

Alternatively, another way to check is by converting each group as I did before. So yes, the answer is 2516 in octal.

But let me just confirm once again by converting $2516_8$ back to binary to see if it's $10101001110_2$.

2516 in octal:

Each digit corresponds to three binary digits:

$2 \rightarrow 010$
$5 \rightarrow 101$
$1 \rightarrow 001$
$6 \rightarrow 110$

So putting them together: 010 101 001 110 $\rightarrow$ 010101001110, which is the original number. So that's correct.

Therefore, the final answer is \boxed{2516} in base eight.

## B.3 Demonstration of SPRINT parallelism within a single solution method

The sample in Section B.2 shows SPRINT parallelizing alternative solution methods in stages 1 and 2. In the below example, we show that SPRINT can also split a single solution method into subtasks, execute them in parallel, and aggregate the results later. Here, the planner splits the problem into calculating the vertical asymptote and re-evaluating the horizontal asymptote, as prompts 1.1 and 1.2 in parallel.

**Question:** The graph of $\frac{2x}{x^2-5x-14}$ has vertical asymptotes $x = a$ and $x = b$, and horizontal asymptote $y = c$. Find $a + b + c$.

---

**Trajectory sample for parallelism within a single solution method**

```
<Plan_1>
```

Determine the vertical and horizontal asymptotes of f(x)=\frac{2x}{x^2-5x-14}, then compute a+b+c.

**`<prompt_1.1>` Identify the vertical asymptotes by finding the roots of the denominator. `</prompt_1.1>`**

**`<prompt_1.2>` Re-evaluate the horizontal asymptote using the degrees of the numerator and denominator. `</prompt_1.2>`**

```
</Plan_1>

<Execution_1>
```

```
<execution_1.1>
```
Vertical asymptotes occur where the denominator is zero (provided the numerator isn't zero). Factoring x^2-5x-14 = (x-7)(x+2) gives roots at x=7 and x=-2. Thus, a=7 and b=-2.
```
</execution_1.1>
```

```
<execution_1.2>
```
Since the numerator's degree (1) is less than the denominator's (2), the horizontal asymptote is y=0. Hence, c=0.
```
</execution_1.2>

</Execution_1>
```

Rest of the solution continues...

---

## B.4 Demonstration of SPRINT execution on a non-STEM task

We use the below example to demonstrate that SPRINT generalizes well to non-STEM tasks like creative writing, despite being different from tasks in the MATH dataset used for fine-tuning. When tasked with generating a brief movie scene description, the planner successfully devises three independent subtasks, each executed concurrently by separate executors. After three planning and execution rounds, these independently developed results were integrated into a coherent and unified narrative.

**Trajectory sample for creative writing task**

```
<Plan_1>
```

    The user wants a 250-word scene set shortly after the Battle of King's Landing.

    `<prompt_1.1>` **Identify the key elements and emotions for each perspective: Bran Stark's visions, the noble's skepticism, and the urchin's admiration.** `</prompt_1.1>`

```
</Plan_1>
```

```
<Execution_1>
```

    `<execution_1.1>`
    - Bran: He's the one who saw the vision of the White City...
    - Noble from the Vale: He's skeptical...
    - Urchin: This adds an unexpected element...
    `</execution_1.1>`

```
</Execution_1>
```

```
<Plan_2>
```

    Based on Execution 1.1:
    Bran: His lines should include visions...
    `<prompt_2.1>` **Develop Bran Stark's lines to describe the vision of the White City and the Iron Throne.** `</prompt_2.1>`

    `<prompt_2.2>` **Develop the noble's lines to express skepticism about Bran's leadership and his brother's choice.** `</prompt_2.2>`

    `<prompt_2.3>` **Develop the urchin's lines to show admiration for Bran and the vision, and the impact on his journey.** `</prompt_2.3>`

```
</Plan_2>
```

```
<Execution_2>
```

    `<execution_2.1>`
    The urchin's lines should be simple but full of wonder...
    `</execution_2.1>`

    `<execution_2.2>`
    The urchin in the Red Keep adds a layer of innocence, contrasting with...
    `</execution_2.2>`

    `<execution_2.3>`
    The streets below are in ruins, but the urchin's eyes...
    `</execution_2.3>`

```
</Execution_2>
```

```
<Plan_3>
```

Based on Executions 2.1–2.3:

`<prompt_3.1>` **Ensure each character's perspective is distinct and contributes to the overall mood. The scene should capture the transition from battle chaos to the realm of possibility of a new order.** `</prompt_3.1>`

`</Plan_3>`

`<Execution_3>`

    `<execution_3.1>`
    The urchin's admiration is genuine...
    `</execution_3.1>`

`</Execution_3>`

`<Final_answer>`

The air was thick with the scent of incense and the faint hum of machinery. The throne loomed before Bran...

The council chambers were quiet, save for the faint murmur of leather on wood...

The Red Keep was alive with the promise of change ... The White City was his home, and the Iron Throne was his throne.

`</Final_answer>`

