# OpenReview forum: "SPRINT: Enabling Interleaved Planning and Parallelized Execution in Reasoning Models"
_NeurIPS.cc/2025/Conference — NeurIPS 2025 poster_

### Official Review · Reviewer_76Mf · 2025-06-22

**Clarity:** 3
**Significance:** 3
**Originality:** 3
**Rating:** 4
**Confidence:** 3

**Summary:**

This paper proposes SPRINT, a method to enable LRMs to parallelize reasoning and execution via planning. The proposed SPRINT framework successfully reduces sequential token generation compared to vanilla fine-tuned LLMs or inference-time parallel techniques such as the skeleton of thoughts. Moreover, the model fine-tuned with SPRINT also shows generalization capabilities in other domains (e.g., model fine-tuned on MATH generalizes on Countdown and GPQA-Diamond).

**Questions:**

Please see the weakness section above. On top of the weaknesses, can the authors also clarify why MATH500 test results do not seem to present integers in terms of correct cases? To my understanding, MATH500 has 500 test cases. According to Table 2, SPRINT seems to have 462.5 cases correct with an accuracy of 92.5.

**Ethical Concerns:**

["NO or VERY MINOR ethics concerns only"]

**Final Justification:**

My concerns are resolved.

**Limitations:**

yes

**Paper Formatting Concerns:**

I have no formatting concerns.

**Quality:**

3

**Strengths And Weaknesses:**

Strengths:

1. The topic is important (significance): how to make LLM/LRM inference faster so that users can enjoy better service.

2. The paper is well written and easy to follow (clarity). The figures and tables are well organized and clearly presented.

3. The proposed method is novel (novelty). It reduces sequential token generation without degrading model performance, and its advantage generalizes across domains.

Weaknesses:

1. Lack of significance test (quality): Although the authors claim that SPRINT has higher accuracy in section 4.2, it seems that the margin is quite small compared to RFT in Table 2. Can the authors present statistical significance tests and clarify whether the performance gain is significant?

2. Lack of analysis about pre-filing latency analysis (quality): The main advantage of SPRINT, to my understanding, is that it reduces the wall-clock time of model inference. While the results indeed show that the sequential generation length is lower, it does not necessarily mean that the total generation time/cost is shorter/lower. Note that in every execution, the executor model has to pre-fill the plan generated by the orchestrator, then execute the plan. The pre-filling of SPRINT appears as an overhead compared to RFT in practice, while it neither appears in Figure 3 nor in other analysis parts. Can the authors clarify whether redundant pre-filing would affect SPRINT's advantage?

---

> ### Author Rebuttal · Authors · 2025-07-31
>
> Thank you for your thoughtful and technical review! We are glad you found the topic’s significance, the paper’s clarity, and SPRINT’s novelty and cross-domain generalization compelling.
> In response to your feedback, we ran additional experiments, provided numerical analysis, and clarified points of ambiguity. In particular, we conducted a statistical significance test, measured runtime metrics, and demonstrated the scaled benefits of SPRINT, especially on longer reasoning trajectories.
>
> ---
>
> **W1: Statistical Significance Analysis of the Results**
>
> While the difference in accuracy between SPRINT and RFT may appear small, our primary contribution lies in reducing latency through parallelism **without sacrificing** the accuracy of the underlying reasoning model. To validate that the observed accuracy gains are meaningful, we conducted a statistical significance analysis using 5 independent runs of both SPRINT and RFT on the MATH500 dataset.
>
> SPRINT achieved an average accuracy of 92.36% with a standard deviation of 0.15, while RFT averaged 90.72% with a standard deviation of 0.24. A two-sample t-test yielded a p-value of 0.00024, indicating that the performance difference is statistically significant.
>
> ---
>
> **W2: Quantifying Prefilling Overhead and End-to-End Latency of SPRINT**
>
> We appreciate the reviewer’s point on the need to quantify SPRINT’s prefilling overhead. To address this, we measured SPRINT’s runtime both **excluding** and **including** the Time‑To‑First‑Token (TTFT) overhead (which captures the cost of the “prefilling” phase), and compared these to Reasoning Fine-Tuned (RFT) model’s total inference latency:
>
> | Dataset                     | SPRINT (Decode Only) | SPRINT (End-to-End) | RFT (End-to-End) |
> |-----------------------------|----------------------|----------------------|------------------|
> | MATH500 (all problems)      | 35.87 s              | 36.92 s              | 40.57 s          |
> | MATH500 (RFT > 8k tokens)   | 73.43 s              | 74.47 s              | 120.54 s         |
>
> **Prefill overhead is minor.** On average, each prefilling step adds only ~0.11 s, while decoding a single token takes ~0.01 s. Even with multiple stages, the bulk of runtime remains in the decoding phase.
>
> **End-to-end speedups.** When including TTFT, SPRINT still outperforms RFT by approximately **9%** on the full MATH500 set (36.92 s vs. 40.57 s), and by approximately **38%** on the longest sequences (74.47 s vs. 120.54 s).
>
> **Scaling benefits.** Because decoding cost dominates, **SPRINT’s advantage increases with trace length**—longer reasoning chains yield greater absolute and relative latency reductions.
>
> **In summary**, even when accounting for prefilling overhead, **SPRINT achieves substantial end-to-end latency reductions** compared to RFT, with the gains becoming more pronounced on tasks that involve very long reasoning traces.
>
> ---
>
> **Q1: Averaged over independent runs**
>
> Regarding the reviewer’s concern about the fractional number of correct cases in MATH500, we would like to clarify that the reported accuracy is the average over independent runs of SPRINT. As a result, while each individual run produces an integer count of correct answers, the average can be non-integer.
>
> ---
>
> Once again, we thank you for your valuable time and feedback. We hope that our additional experiments, analysis of results, and discussion addressed your concerns. We hope you find the paper significantly improved and consider reflecting this in your final score.

---

> > ### Author Response · Authors · 2025-08-03
> > **Reminder**
> >
> > We’re writing to kindly follow up on our responses to your valuable comments. If you haven’t had a chance yet, we would greatly appreciate it if you could take a moment to review our responses and, if appropriate, update your scores accordingly.
> >
> > Thank you again for your time and valuable feedback throughout this process. Please don’t hesitate to let us know if any further clarification is needed from our side.

---

> > > ### Comment · Reviewer_76Mf · 2025-08-04
> > >
> > > Thank the authors for the comment. I will keep my score.

---

### Official Review · Reviewer_6CkS · 2025-06-29

**Clarity:** 3
**Significance:** 3
**Originality:** 3
**Rating:** 5
**Confidence:** 3

**Summary:**

This paper presents SPRINT, a post-training and inference-time framework that enables large reasoning models to dynamically identify and execute independent reasoning subtasks in parallel, reducing inference length and time. SPRINT uses a data curation pipeline to restructure reasoning trajectories into rounds of planning and parallel execution, followed by fine-tuning on a small amount of curated data. Experiments on reasoning tasks show that SPRINT reduces the number of sequential tokens on long reasoning chains, while achieving comparable performance to the reasoning fine-tuned model.

**Questions:**

1. Since Table 1 mentions PASTA [1] and Hogwild! Inference [2], and highlights SPRINT’s advantages over them, why not include these methods as quantitative baselines in the experimental comparisons?
2. I agree with the authors’ point in Lines 144–147 that SPRINT could be more valuable for tool-integrated reasoning tasks. So why not directly evaluate SPRINT on such benchmarks, such as mathematical benchmarks with tool calls (those used in [3]), xLAM [4], or BrowseComp [5] with search tools? The rationale for evaluating only on MATH500, GPQA-diamond, and Countdown remains unclear.
3. The source dataset for Figure 5 is unspecified. Does it show results from MATH500 or from the out-of-distribution tasks (GPQA-diamond, Countdown)? Also, does the same performance trend hold in the out-of-distribution scenarios?
4. I recommend adding failure case analysis and a more detailed discussion on possible improvements. Specifically, it would help to clarify whether most errors stem from subtask planning or from execution failures during parallelization.
5. The sole reliance on Seq. Tokens as the efficiency metric seems insufficient. Although the authors note that "due to limited resources, we were unable to implement the optimal hardware-accelerated decoding for SPRINT," including other runtime metrics would significantly strengthen the paper’s efficiency claims.

[1] Tian Jin, Ellie Y. Cheng, Zack Ankner, Nikunj Saunshi, Blake M. Elias, Amir Yazdanbakhsh, Jonathan Ragan-Kelley, Suvinay Subramanian, and Michael Carbin. Learning to keep a promise: Scaling language model decoding parallelism with learned asynchronous decoding, 2025.

[2] Gleb Rodionov, Roman Garipov, Alina Shutova, George Yakushev, Vage Egiazarian, Anton Sinitsin, Denis Kuznedelev, and Dan Alistarh. Hogwild! inference: Parallel llm generation via concurrent attention, 2025.

[3] Li, Xuefeng, Haoyang Zou, and Pengfei Liu. Torl: Scaling tool-integrated rl. arXiv preprint arXiv:2503.23383.

[4] Jianguo Zhang, Tian Lan, Ming Zhu, Zuxin Liu, Thai Hoang, Shirley Kokane, Weiran Yao, Juntao Tan, Akshara Prabhakar, Haolin Chen, et al. 2024. xlam: A family of large action models to empower ai agent systems. arXiv preprint arXiv:2409.03215.

[5] Wei, Jason, et al. Browsecomp: A simple yet challenging benchmark for browsing agents. arXiv preprint arXiv:2504.12516.

**Ethical Concerns:**

["NO or VERY MINOR ethics concerns only"]

**Final Justification:**

I have carefully reviewed the authors’ detailed responses to my 5 questions. The additional experiment results and clarifications adequately resolve my earlier concerns, particularly regarding Q3, Q4, and Q5. Adding runtime as an additional efficiency metric can offer stronger evidence for the paper’s contributions. Therefore, I raise my score to 5.

**Limitations:**

Yes

**Paper Formatting Concerns:**

No concerns

**Quality:**

3

**Strengths And Weaknesses:**

Strengths:
1. The motivation is clear and well-justified. The core idea—using curated data to fine-tune LLMs for dynamically identifying independent subtasks within extended reasoning processes and executing them in parallel—is simple and effective. The methodology section is well-written and easy-to-follow.
2. The supplementary material is extensive and informative, providing details for reproduction.
3. As discussed in the future work section, this approach has promising potential for extension to latency-aware reinforcement learning.

Weaknesses:
1. PASTA and Hogwild! Inference are mentioned but not included as baselines in experiments. (Q1)
2. Why not directly evaluate SPRINT on tool-integrated reasoning benchmarks to better demostrate its efficiency? (Q2)
3. The source dataset for Figure 5 is unspecified, and it’s unclear if the observed trends hold for out-of-distribution tasks. (Q3)
4. I recommend adding failure case analysis on where SPRINT fails. (Q4)
5. The use of only Seq. Tokens as an efficiency measure is limited. (Q5)

---

> ### Author Rebuttal · Authors · 2025-07-31
>
> Thank you for your detailed review and thoughtful suggestions! We appreciate your acknowledgement of SPRINT’s clear motivation, effective core idea, and its great potential for extension to latency-aware reinforcement learning.
>
> To address your comments, we conducted additional experiments, clarified points of ambiguity, and provided further analysis. These include evaluating additional baselines such as Hogwild! Inference with two models, measuring runtime metrics for both SPRINT and the competing baseline, and offering a detailed analysis of failure cases along with a discussion of potential improvements.
>
> ---
>
> **Q1/W1: Hogwild! Inference and PASTA as Baselines**
>
> We have evaluated the performance of **Hogwild! Inference** on MATH500 using two types of base models: a chat-instruct model (**Qwen-2.5-7B-instruct**) and a reasoning model (**DeepSeek-R1-Distill-Qwen-7B**). As shown in the table below, both configurations result in substantially lower accuracy compared to SPRINT, which achieves **92.5%** accuracy. We attribute this to the fact that Hogwild! Inference does not involve fine-tuning for the planning-and-parallel-execution paradigm. This is consistent with the authors' own observations, as they also report subpar performance when applying Hogwild! to reasoning-focused models.
>
> | Base Model     | Accuracy (%) | Seq Tokens | Total Tokens |
> |----------------|--------------|------------|--------------|
> | Chat-Instruct  | 61.0         | 1173       | 2346         |
> | Reasoning      | 63.8         | 1797       | 3595         |
>
> Regarding **PASTA**, we have not included it as a baseline because the authors have not released their **code** or **trained model**, which prevents direct reproduction or comparison. Additionally, PASTA was not evaluated on the same benchmarks we use for SPRINT. Their experiments focus on simpler tasks that do not require complex, multi-step reasoning.
>
> PASTA decomposes a problem into subtasks that are executed in parallel **within a single stage**, and then aggregates the results to form a final answer. In contrast, **SPRINT** performs planning and execution **over multiple stages**, which enables more sophisticated, iterative decomposition and recomposition of reasoning steps.
>
> Notably, the **SoT-chat** and **SoT-reasoning** baselines help illustrate this difference, showing how chat-instruct and reasoning models, when not fine-tuned for the planning-and-parallel-execution setup, perform when only a single round of planning and execution is allowed.
>
> ---
>
> **Q2/W2: Tool-integrated reasoning**
>
> We thank the reviewer for highlighting the potential of SPRINT in tool-integrated reasoning scenarios. That said, evaluating SPRINT on tool-augmented benchmarks such as those used in xLAM, or BrowseComp presents a distinct challenge. Specifically, the open-source base models used in our work (e.g., the DeepSeek-R1-distilled series) lack native tool-calling capabilities, including the ability to produce correctly formatted tool-specific outputs. Incorporating such capabilities would require substantial additional fine-tuning, which is orthogonal to the scope of our current investigation. On the other hand, by evaluating models without tool integration, we allow a direct comparison against the baseline reasoning models that are also not tool-integrated. Otherwise, the effects on latency and accuracy resulting from parallelization and tool-use could get conflated.
> The main objective of our work is to demonstrate that it is algorithmically feasible to train reasoning models to perform interleaved planning and parallel execution without relying on handcrafted pipelines or static decomposition strategies. Our contribution thus focuses on enabling model-driven, adaptive parallelization in complex reasoning tasks. Looking ahead, we view the integration of tool-calling capabilities with SPRINT’s parallel planning mechanism as a promising direction. Equipping models with both capabilities could enable parallel tool-use, expanding SPRINT’s applicability to tool-augmented reasoning tasks.
>
> ---
>
> **Q3/W3: Clarification on Figure 5 Dataset and Parallelization Trends in Out-of-Distribution Tasks**
> Thank you for bringing this to our attention. Figure 5 is based on the logs from MATH500 using the dataset’s original difficulty levels. To address your question regarding the out-of-distribution (OOD) tasks (GPQA-diamond and Countdown): yes, we observe a similar performance trend. Specifically, more parallelization opportunities arise in the earlier stages, where the model explores alternative strategies and identifies subtasks before converging to the final solution.
>
>
> The detailed results are as follows:
>
> **GPQA**
> - Number of problems reaching stage *i*:
>   [198, 185, 157, 114, 81, 50, 35, 26, 18, 13, 8]
> - Number of problems exhibiting parallelism (more than one execution) at stage *i*:
>   [91, 100, 65, 44, 24, 7, 3, 5, 2, 0, 0]
>
> **Countdown**
> - Number of problems reaching stage *i*:
>   [1000, 991, 886, 732, 498, 382, 275, 202, 150, 124, 72]
> - Number of problems exhibiting parallelism (more than one execution) at stage *i*:
>   [585, 655, 340, 176, 102, 48, 29, 13, 8, 4, 4]
>
> Due to rebuttal constraints, we are unable to upload a separate file or figure at this stage. However, we will include the corresponding analysis figure for the OOD benchmarks in the final version of the paper.
>
> ---
>
> **Q4/W4: Adding failure case analysis**
> Following the reviewer’s recommendation, we have analyzed failure cases and identified three categories:
> Logical Errors During Execution: These include calculation mistakes, typographical errors, or incorrect mathematical assumptions.
> Improper Conflict Resolution: These occur when two executors independently pursue different solution paths and produce conflicting results, causing the planner to fail in selecting the correct one.
> Incomplete Plan/Execution: In some complex tasks, the model repeatedly tries various approaches but fails to arrive at a final answer within the maximum number of permitted stages (12 stages). In other cases, the execution within a stage becomes too lengthy and gets truncated when it reaches the maximum token count.
> The logical errors during execution largely stem from the limitations of the base model, rather than from parallelization or planning mechanisms. In fact, similar types of errors are also observed in the baseline Reasoning Fine-tuned (RFT) model.
> Based on the scaling trends of the reasoning models, we expect that with larger and more capable base models, both execution and planning quality will improve. We supervised fine-tuned our model on fewer than 2,000 data points, which successfully unlocked the ability to perform interleaved planning and parallel execution. Future work can enhance the model’s performance further by training on larger datasets and incorporating reinforcement learning.
> We will include a detailed failure analysis, along with illustrative examples and proposed directions for improvement, in the revised version of the paper.
>
> ---
>
> **Q5/W5: Runtime as an Efficiency Metric**
>
> We agree that reporting only the number of generated tokens does not fully capture end‑to‑end efficiency. To strengthen our claims, we measured **SPRINT’s runtime** both **excluding** and **including** the Time‑To‑First‑Token (TTFT) overhead, which accounts for the cost of the "prefilling" phase, and compared these to the **Reasoning Fine-Tuned (RFT)** model's total inference latency:
>
> | Dataset                    | SPRINT (Decode Only) | SPRINT (End-to-End) | RFT (End-to-End) |
> |----------------------------|----------------------|----------------------|------------------|
> | MATH500 (all problems)     | 35.87 s              | 36.92 s              | 40.57 s          |
> | MATH500 (RFT > 8k tokens)  | 73.43 s              | 74.47 s              | 120.54 s         |
>
> **Prefill overhead is minor.** On average, each prefilling step adds only ~0.11 s, while decoding a single token takes ~0.01 s. Even across multiple stages, the bulk of runtime remains in the decoding phase.
>
> **End‑to‑end speedups.** When including TTFT, SPRINT still outperforms RFT by approximately **9%** on the full MATH500 set (36.92 s vs. 40.57 s), and by approximately **38%** on the longest sequences (74.47 s vs. 120.54 s).
>
> **Scaling benefits.** Because decoding cost dominates, **SPRINT’s advantage grows with trajectory length**: longer reasoning chains yield both larger absolute and relative latency reductions.
>
> **In summary**, even accounting for prefilling overhead, SPRINT achieves substantial end‑to‑end latency reductions compared to RFT, and these gains increase significantly for tasks that involve long reasoning trajectories.
>
> ---
>
> We appreciate the comprehensive comments and hope that our additional experiments, analysis of results, and discussion addressed your concerns. We hope you find the paper significantly improved and consider reflecting this in your final score.

---

> > ### Comment · Reviewer_6CkS · 2025-08-02
> > **Thanks for the Detailed Response**
> >
> > I have carefully reviewed the authors’ detailed responses to my 5 questions. The additional experiment results and clarifications adequately resolve my earlier concerns, particularly regarding Q3, Q4, and Q5. Adding runtime as an additional efficiency metric can offer stronger evidence for the paper’s contributions. I will update my score during the upcoming Reviewer-AC Discussion period.

---

> > > ### Author Response · Authors · 2025-08-03
> > > **Thank you for your feedback and consideration**
> > >
> > > Thank you for your thoughtful review. We’re grateful that our clarifications addressed your concerns, and we appreciate your plan to raise the score during the discussion period.

---

### Official Review · Reviewer_tuC4 · 2025-07-02

**Clarity:** 3
**Significance:** 2
**Originality:** 3
**Rating:** 4
**Confidence:** 3

**Summary:**

This paper introduces SPRINT, a method that enables models to parallelize reasoning and execution processes through post-training and inference-time techniques. SPRINT incorporates a multi-stage data curation pipeline that transforms sequential reasoning processes into structured multi-step formats. Using the curated data, the authors conduct experiments on the MATH dataset and OOD datasets such as GPQA and Countdown. The results show a reduction in sequential tokens while maintaining/improving accuracy.

**Questions:**

1. Can you provide an analysis of the total number of tokens used, including both sequential and parallel branches?
2. What is the impact of the number of parallel branches within a reasoning step? Does increasing the number of parallel branches consistently improve performance, or are there any trade-offs?
3. Can you offer further insights into why parallelization works in this context? For example, do parallel steps offer diverse solution paths, or are they complementary substeps that can be merged into a final answer?

**Ethical Concerns:**

["NO or VERY MINOR ethics concerns only"]

**Final Justification:**

The author responses have addressed my main concerns regarding the total token usage and implementation details, although the performance on MATH appears to be saturated. The explanation for why the parallelization helps also makes sense to me. Therefore, I have increased my score.

**Limitations:**

Yes.

**Paper Formatting Concerns:**

NA.

**Quality:**

3

**Strengths And Weaknesses:**

Strengths:
1. This work introduces dynamic parallelism in reasoning processes and proposes data creation strategies to enable such parallelization.
2. The analysis is interesting, providing comparisons across problem difficulty levels and reasoning lengths.
3. The overall writing is clear and easy to follow.

Weaknesses:
1. The evaluation of efficiency primarily relies on the count of sequential tokens, which may not fully reflect the total computational cost. A more complete analysis should include the total token usage, including tokens used in parallel branches.
2. The paper lacks several design and implementation details, such as concrete examples of parallelization, analysis of steps versus stages, and specifics regarding decoding hyperparameters used during inference.

---

> ### Author Rebuttal · Authors · 2025-07-31
>
> We thank the reviewer tuC4 for your constructive feedback! We appreciate your recognition of our contributions, particularly the introduction of dynamic parallelism in reasoning processes, our proposed data creation strategies to support such parallelization, and the analysis across problem difficulty levels and reasoning lengths.
>
> In response to your comments, we have provided further numerical analysis, illustrative examples, and discussion to address the points you raised. These include clarifications on implementation details, token counts, and the characteristics of SPRINT’s dynamic parallelism. We plan to incorporate these improvements into the final version of the paper and hope you will consider updating your score based on these additions.
>
>
> **W1/Q1: Efficiency metrics, total token usage**
> Following your suggestion, we measured the total number of tokens generated by SPRINT and other baselines, along with the two other metrics of accuracy and #seq tokens averaged over the MATH500 dataset.
>
> | Method                  | Accuracy (%) | Sequential Tokens | Total Tokens |
> |-------------------------|--------------|-------------------|--------------|
> | Self-consistency        | 80.6         | 590               | 11,645       |
> | SoT-chat                | 47.3         | 256               | 1,290        |
> | SoT-reasoning           | 90.8         | 3,836             | 11,538       |
> | Reasoning Fine-tuned (RFT) | 91.0     | 2,880             | 2,880        |
> | **SPRINT**              | **92.5**     | **2,440**         | **3,622**    |
>
> Among the high-performing methods (accuracy > 90%), SoT-reasoning exhibits a high total token cost due to redundant work by multiple independent threads, which individually replicate similar computations without coordination. In contrast, SPRINT achieves better efficiency by enabling coordination between parallel executors and the planner, resulting in reduced redundant computation. Although SPRINT introduces some planning and synchronization overhead, it substantially reduces **sequential tokens**, the key bottleneck for latency, relative to RFT, while improving accuracy and keeping total tokens low.
>
>
> **W2: Clarification on the Design and Implementation Details**
>
> We have provided the design and implementation details in the file **Appendix-SPRINT.pdf**, submitted as part of our **supplementary materials**. In particular, inference-time hyperparameters, such as temperature (0.6), maximum tokens (36,000), and others, are described in Appendix A.1 (Implementation Details).
> The analysis of the number of parallel plans generated per stage is presented in **Figure 5** and discussed at the end of **Section 4.2**. We show that SPRINT dynamically adjusts the number of parallel plans per stage: in the early stages, the planner proposes more plans to explore different pathways and tackle the subtasks, and as the solution progresses, the planner proposes fewer plans to aggregate the results.
> Moreover, we include concrete examples of parallelization in **Appendix B (Sample Demonstrations)**, and we provide additional examples of other forms of parallelization within this rebuttal (see above and below).
>
> **Q2: Impact of the Number of Parallel Branches per Reasoning Step**
>
> We would like to clarify that the number of parallel branches at each reasoning step is **not** a manually tuned hyperparameter in our system. Instead, SPRINT’s planner is trained to **dynamically propose** the number of parallel plans to be executed at each step based on the problem context and solution state.
>
> Through the post-training process, the planner learns to identify parallelization opportunities and assign subtasks accordingly. As a result:
> - In early stages, it proposes more plans to explore alternative reasoning pathways and tackle subtasks in parallel.
> - In later stages, it generates fewer plans, focusing on aggregating partial results and converging to a final answer.
> - For harder problems, the planner tends to propose more parallel branches compared to easier problems.
>
> Both of these trends are evident in **Figure 5** (MATH500), where we observe stage-wise and difficulty-based adaptation in parallelization behavior.
>
>
> **Q3: Why Does Parallelization Work in SPRINT?**
> Parallelization can be effective in three main ways:
> 1. **Exploring alternative solutions**
> 2. **Solving complementary substeps**
> 3. **Verifying a previous step while another task progresses**
>
> Below, we discuss two examples of the SPRINT’s reasoning trajectories illustrating each of these mechanisms.
>
> ---
>
> **Exploring Alternative Solutions and Verification**
> In the trajectory presented in **Appendix B.2**, the task involves converting a binary number into its base-eight representation. SPRINT’s planner suggests two alternative approaches in parallel:
> - (a) Convert the binary number to decimal and then to octal.
> - (b) Split the 12-digit binary number into groups of three digits and convert each group directly.
>
> Both paths proceed in parallel, and then at **Stage 3**, the planner issues a verification prompt while finalizing the answer in parallel:
> - **Prompt 3.1**: Verify the correctness of the grouping-based conversion (solution b).
> - **Prompt 3.2**: Reflect on the correctness of the final answer by comparing the outcomes of both conversion methods.
>
> This structure helps the system reach a reliable solution through internal consistency checks, enhancing robustness in ambiguous cases.
>
> ---
>
> **Decomposing into Complementary Subtasks**
> In another example, the planner splits the problem into two **complementary subtasks** in parallel: calculating vertical and horizontal asymptotes.
>
> **Question**
> *The graph of* \\(f(x)=\frac{2x}{x^2-5x-14}\\) *has vertical asymptotes* \\(x=a\\) *and* \\(x=b\\), *and horizontal asymptote* \\(y=c\\). *Find* \\(a+b+c\\).
>
> ---
>
> **Plan 1**
> To solve the problem, we need to:
> - Identify the vertical asymptotes by finding the roots of the denominator.
>   - **Prompt 1.1**: Identify the vertical asymptotes by finding the roots of the denominator of the function.
> - Re-evaluate the horizontal asymptote based on degree comparison.
>   - **Prompt 1.2**: Re-evaluate the horizontal asymptote using the degrees of the numerator and denominator.
>
> ---
>
> **Execution 1**
>
> - **Exec 1.1**
>   Vertical asymptotes occur where the denominator is zero (as long as the numerator isn’t).
>   Factor \\(x^2 - 5x - 14 = (x - 7)(x + 2)\\), so the vertical asymptotes are at \\(x = 7\\) and \\(x = -2\\).
>   Therefore, \\(a = 7\\), \\(b = -2\\).
>
> - **Exec 1.2**
>   Since the degree of the numerator is less than the denominator, the horizontal asymptote is \\(y = 0\\), so \\(c = 0\\).
>
> ---
>
> These parallel subtasks are merged to compute the final result:
> \\[
> a + b + c = 7 + (-2) + 0 = \boxed{5}
> \\]
>
> This example illustrates how SPRINT can decompose and parallelize logically independent subtasks that are later recombined to yield the final answer.
>
> The full log can be found in our repository at “logs_full_trajectory/SPRINT/MATH-500_default/planner_trajectory_test_algebra_621.txt”.
>
> Once again, we thank you for your valuable time and valuable feedback. We hope that the additional numerical analysis, illustrative examples, and clarifications we have provided address your concerns. We believe these improvements have strengthened the paper and hope you will consider reflecting this in your final score.

---

> > ### Author Response · Authors · 2025-08-03
> > **Reminder**
> >
> > We’re writing to kindly follow up on our responses to your valuable comments. If you haven’t had a chance yet, we would greatly appreciate it if you could take a moment to review our responses and, if appropriate, update your scores accordingly.
> >
> > Thank you again for your time and valuable feedback throughout this process. Please don’t hesitate to let us know if any further clarification is needed from our side.

---

> > ### Comment · Reviewer_tuC4 · 2025-08-05
> >
> > Thank you for the detailed response and additional analysis. I encourage the authors to conduct the total token analysis to other datasets as well, as performance on MATH appears to be saturated. Aside from that, most of my concerns have been addressed. I have raised my score accordingly.

---

> > > ### Author Response · Authors · 2025-08-08
> > > **Response to Reviewer Feedback**
> > >
> > > Thank you for your thoughtful feedback and for raising your score. We’re glad our rebuttal has addressed your concerns, and we will ensure to incorporate your feedback into the updated version of our paper.

---

### Official Review · Reviewer_5uvg · 2025-07-02

**Clarity:** 3
**Significance:** 3
**Originality:** 3
**Rating:** 5
**Confidence:** 4

**Summary:**

This paper uses SFT training LRMs to automatically combine traditional sequential reasoning with parallel task decomposition and execution, aiming at improvement of both accuracy and efficiency.

**Questions:**

- Q1: The examples in the appendix seem to show the tendency of LRMs using parallelism for different reasoning and soltion paths. Are there other cases that the model set plans for parallelizable sub-tasks of one single solution?
- Q2: What the recovery behavior of LRMs in SPRINT is like? Take the appendix figure as an example, what the LRM would usually do if an error was detected in the rechecking execution?

**Ethical Concerns:**

["NO or VERY MINOR ethics concerns only"]

**Final Justification:**

Raising rating for new experiments on creative tasks and the clarified notions.

**Limitations:**

yes

**Quality:**

2

**Strengths And Weaknesses:**

### **Strengths**

#### **S1: Reasonable Motivation**

The idea to parallelize reasoning and execution process makes a lot sense to me. The concept of combining sequential and parallel actions brings a novel aspect of investigating LRMs.

#### **S2: Potentially Very Useful on Certain Tasks**

On tasks with clear sub-task splits or natural parallelism properties, the proposed techniques can be very useful in terms of both accuracy and efficiency.

### **Weaknesses**

#### **W1: Issues about Efficiency Evaluation**

Firstly, I think the "number of sequencial tokens" defined in Section 4.1 underestimates the cost of SPRINT. Since all the plannings are sequencially generated by the head agent, to calculate the token consumption related to time cost estimation, it should be defined as:

$ \sum_{i=1}^{stages} \max_k^{prompts} [(\sum_{j=1}^k P_{i, j}) +E_{i,k}]. $

If defined this way, it is possible that SPRINT turns out decreasing the efficiency rather than increasing it.

Secondly, other than sequencial tokens, it may help report other two types of token statistics:
- Number of memorized tokens, defined as all the tokens whoses kv caches need to be stored in the head agent, useful for comparing memory use;
- Number of all tokens, defined as all the tokens generated by both the head and the works, useful for comparing computation consumption or API cost.

#### **W2: Generalizability Concerns**

The experiments are conducted on STEM tasks which are easier to be splitted in to sub-goals. However, for other tasks such as common sense reasoning and creative writing, the proposed technique might be less useful.

---

> ### Author Rebuttal · Authors · 2025-07-31
>
> We thank the reviewer 5uvg for your thoughtful review! We appreciate the reviewer’s recognition of the novelty of SPRINT in bringing parallelism in large reasoning models and its high utility in terms of accuracy and efficiency.
>
> In response to your comments, we provided the numerical analysis, discussions, and illustrative examples on the SPRINT’s generalization, sub-task decomposition, and error recovery behaviour. We plan to incorporate these additions into the final version of the paper and hope you will consider updating your score in light of these improvements.
>
> **W1.1: Efficiency Evaluation**
>
> We wish to clarify a potential misunderstanding about our notations and evaluation metric for the “number of sequential tokens” as defined in Section 4.1.
>
> In our paper, **\\(P_{i,j}\\)** is the **cumulative** token count generated by the planner from the start of stage i up to and including prompt j. As soon as the planner streams the final token of prompt j, its corresponding execution can begin, without waiting for later prompts.
>
> Accordingly, the sequential‐token cost at stage i is correctly given by
> \\[
> \max_{k}\bigl[P_{i,k} + E_{i,k}\bigr],
> \\]
> and the total cost sums over stages:
> \\[
> \sum_{i=1}^{\text{stages}} \max_{k}\bigl[P_{i,k} + E_{i,k}\bigr].
> \\]
>
> The reviewer’s suggested formula assumes \\(P_{i,j}\\) is **per-prompt** rather than cumulative. Since our \\(P_{i,j}\\) already accumulates all prior tokens, \\(\sum_{j=1}^{k}P_{i,j}=P_{i,k}\\). Thus, both expressions coincide under our definitions.
>
> **W1.2: Total and memorized tokens**
>
> Following your suggestion, we measured and report two additional token statistics:
>
> **Memorized Tokens**
>
> During the execution of a given stage, the KV cache contains tokens from all plans (up to and including the current stage) and all executions (up to the current stage). Hence, the number of memorized tokens increases as the stages progress in SPRINT’s solution. The peak usage of the KV cache occurs when the final answer is being generated. We found this peak usage to be 3475 tokens on average over MATH500.
>
> **Total Tokens**
>
> We present these values along with accuracy and sequential token counts in the table below, averaged over the MATH500 dataset:
>
> | Method                  | Accuracy (%) | Sequential Tokens | Total Tokens |
> |-------------------------|--------------|-------------------|--------------|
> | Self-consistency        | 80.6         | 590               | 11,645       |
> | SoT-chat                | 47.3         | 256               | 1,290        |
> | SoT-reasoning           | 90.8         | 3,836             | 11,538       |
> | Reasoning Fine-tuned (RFT) | 91.0     | 2,880             | 2,880        |
> | **SPRINT**              | **92.5**     | **2,440**         | **3,622**    |
>
> Among the high-performing methods (accuracy > 90%), SoT-reasoning exhibits a high total token cost due to redundant work by multiple independent threads, which individually replicate similar computations without coordination. In contrast, SPRINT achieves better efficiency by enabling coordination between parallel executors and the planner, resulting in reduced redundant computation. Although SPRINT introduces some planning and synchronization overhead, it substantially reduces **sequential tokens**, the key bottleneck for latency, relative to RFT, while improving accuracy and keeping total tokens low.
>
> **W2: Generalization of SPRINT beyond the STEM tasks**
>
> We acknowledge the reviewer’s observation that STEM tasks likely benefit more readily from decomposition into subgoals. Indeed, one of our primary motivations in designing SPRINT was to identify and leverage parallelization opportunities to efficiently reduce lengthy sequential CoTs generated for *complex tasks*.
>
> To address the reviewer’s concern, we also evaluated SPRINT on a creative writing task. Interestingly, we observed that even though the model had only been trained on mathematical data, a domain completely different from creative writing, it effectively proposed parallelizable plans when tasked with generating a brief movie scene description.
>
> In the example provided below, at Stage 2, the planner successfully devised three independent subtasks, each executed concurrently by separate executors. After three planning and execution rounds, these independently developed subtasks were integrated into a coherent and unified narrative.
>
> ---
>
> **Query**
> *Write a 250‑word scene set shortly after the Battle of King's Landing, describing the mood in the throne room from different perspectives: Bran Stark, now seated on the Iron Throne, observing the aftermath with his visions. A skeptical noble from the Vale, questioning the new king's rule. A street urchin who snuck into the Red Keep, marveling at the scene.*
>
> ---
>
> **Stage 1: Initial Planning**
>
> - **Plan 1**: The user wants a 250-word scene set shortly after the Battle of King's Landing.
>   - **Prompt 1.1**: Identify the key elements and emotions for each perspective: Bran Stark's visions, the noble's skepticism, and the urchin's admiration.
>
> **Stage 1: Execution**
>
> - **Execution 1.1**:
>   - Bran: He’s the one who saw the vision of the White City...
>   - Noble from the Vale: He's skeptical...
>   - Urchin: This adds an unexpected element...
>
> ---
>
> **Stage 2: Refined Planning**
>
> - Based on Execution 1.1:
>   - Bran: His lines should include visions...
>     - **Prompt 2.1**: Develop Bran Stark's lines to describe the vision of the White City and the Iron Throne.
>     - **Prompt 2.2**: Develop the noble's lines to express skepticism about Bran's leadership and his brother's choice.
>     - **Prompt 2.3**: Develop the urchin's lines to show admiration for Bran and the vision, and the impact on his journey.
>
> **Stage 2: Execution**
>
> - **Exec 2.1**: The urchin's lines should be simple but full of wonder…
> - **Exec 2.2**: The urchin in the Red Keep adds a layer of innocence, contrasting with…
> - **Exec 2.3**: The streets below are in ruins, but the urchin's eyes…
>
> ---
>
> **Stage 3: Final Planning**
>
> - Based on Executions 2.1–2.3:
>   - **Prompt 3.1**: Ensure each character's perspective is distinct and contributes to the overall mood. The scene should capture the transition from battle chaos to the realm of possibility of a new order.
>
> **Stage 3: Execution**
>
> - **Execution 3.1**: The urchin's admiration is genuine…
>
> ---
>
> **Final Answer: The Iron Throne Room**
>
> The air was thick with the scent of incense and the faint hum of machinery. The throne loomed before Bran …
>
> The council chambers were quiet, save for the faint murmur of leather on wood …
>
> The Red Keep was alive with the promise of change … The White City was his home, and the Iron Throne was his throne.
>
> ---
>
> This example illustrates that SPRINT can generalize to creative domains and leverage parallel planning, even outside its training domain. We believe this demonstrates the broader applicability of our method beyond STEM reasoning.
>
> **Q1: Does SPRINT parallelize across sub-tasks?**
>
> Yes, according to the task at hand, the planner might propose plans for trying alternative solution paths, or decomposing a solution into its subtasks to be processed in parallel. The following is an example taken from SPRINT execution, where the planner splits the problem into calculating the vertical asymptote and re-evaluating the horizontal asymptote, as prompts 1 and 2 in parallel.
>
> ---
>
> **Question (algebra_305, MATH500)**
> *The graph of* \\(f(x) = \frac{2x}{x^2 - 5x - 14}\\) *has vertical asymptotes* x=a *and* x=b, *and horizontal asymptote* \\(y=c\\). *Find* \\(a+b+c\\).
>
> ---
>
> **Stage 1: Initial Planning**
>
> - **Plan 1**: Determine the vertical and horizontal asymptotes of \(f(x)=\frac{2x}{x^2-5x-14}\), then compute \(a+b+c\).
>   - **Prompt 1.1**: Identify the vertical asymptotes by finding the roots of the denominator.
>   - **Prompt 1.2**: Re-evaluate the horizontal asymptote using the degrees of the numerator and denominator.
>
> **Stage 1: Execution**
>
> - **Execution 1.1**:
>   Vertical asymptotes occur where the denominator is zero (provided the numerator isn’t zero). Factoring \(x^2-5x-14 = (x-7)(x+2)\) gives roots at \(x=7\) and \(x=-2\). Thus, \(a=7\) and \(b=-2\).
>
> - **Execution 1.2**:
>   Since the numerator’s degree (1) is less than the denominator’s (2), the horizontal asymptote is \(y=0\). Hence, \(c=0\).
>
> ---
>
> *The next planning and execution stages would be followed.*
>
> **Q2: What is the error recovery behavior of SPRINT like?**
>
> We appreciate the reviewer’s insightful question! When an error is detected during execution, **SPRINT** typically follows a structured recovery process:
> 1. **Error Detected** – Identify the mistake during execution.
> 2. **Replanning** – Generate targeted prompts to correct the error.
> 3. **Re-execution** – Run corrected subtasks (often in parallel).
> 4. **Verification** – Validate the final solution against the original problem.
>
> We illustrate SPRINT’s error recovery behavior through a representative thinking trajectory example:
>
> ---
>
> **Question**
> *Convert the quadratic* \\(x^2 + 2.6x + 3.6\\) *into the form* \\((x+b)^2 + c\\), *then find* \\(b + c\\).
>
> ---
>
> **Error in execution**
> - **Execution 2.2 (Initial Error):**
>   An arithmetic mistake led to \\(c = 0.05\\), producing an incorrect sum \\(b + c = 1.74\\).
>
> ---
>
> **Error Detection & Replanning**
> - **Plan 3 (Error Detection):**
>   Issue targeted prompts to explicitly recheck the arithmetic and re-evaluate prior steps.
>
> ---
>
> **Recovery Execution**
> - **Execution 3.1 & 3.2 (Recovery):**
>   Correct arithmetic recalculation finds \\(c = 1.91\\), updating the sum to \\(b + c = 3.21\\).
>
> ---
>
> **Final Verification**
> - **Plan 4 (Verification):**
>   Expand \\((x+b)^2 + c\\) and compare it to the original quadratic to confirm accuracy.
>
> ---
>
> **Final Answer**
> \\[
> b + c = \boxed{3.21}
> \\]
>
> ---
>
> The full log can be found in our repository at “logs_full_trajectory/SPRINT/MATH-500_default/planner_trajectory_test_algebra_621.txt”.

---

> > ### Author Response · Authors · 2025-08-03
> > **Reminder**
> >
> > We’re writing to kindly follow up on our responses to your valuable comments. If you haven’t had a chance yet, we would greatly appreciate it if you could take a moment to review our responses and, if appropriate, update your scores accordingly.
> >
> > Thank you again for your time and valuable feedback throughout this process. Please don’t hesitate to let us know if any further clarification is needed from our side.

---

> > ### Comment · Reviewer_5uvg · 2025-08-05
> > **Thank you**
> >
> > Thank you for your diligent efforts and insightful work! Your response has provided me with new information. I’m increasing my rating from 4 to 5 and my confidence from 3 to 4.

---

> > > ### Author Response · Authors · 2025-08-08
> > > **Response to Reviewer Feedback**
> > >
> > > Thank you for your thoughtful feedback and for taking the time to review our work. We truly appreciate your updated rating and confidence!

---

### Note · Authors · 2025-08-15

We thank the AC and all reviewers for their insightful questions and constructive feedback, which have significantly strengthened our work. We are grateful that our clarifications, additional experiments, and analyses effectively addressed the reviewers’ concerns, leading Reviewers 5uvg, tuC4, and 76Mf to raise their scores.

**Key Contributions & Strengths**

 SPRINT introduces adaptive, model-driven parallelization for multi-step reasoning, reducing sequential tokens and runtime while maintaining or improving accuracy on both in-domain and out-of-domain tasks. Reviewers highlighted its novelty, efficiency–performance gains, robust cross-domain behavior, and clear methodological presentation.

**Enhancements in Response to Feedback**

- **Efficiency & Token Metrics:** We have clarified the definition of “sequential tokens” and added total token counts as an additional metric for comparison. On MATH500, SPRINT reduces the sequential token count by 15% vs. RFT while increasing accuracy and keeping total tokens low.

- **Why Parallelization Works:** We have provided concrete trajectory samples illustrating the benefits of parallelization, specifically how SPRINT performs exploration, complementary subtask decomposition, verification, and error recovery in parallel.

- **Baselines & Comparisons:** We have evaluated Hogwild! inference with two model types and shown that it underperforms SPRNT by a substantial margin (≤63.8% vs. 92.5%). We have also clarified that PASTA has been excluded due to the unavailability of code/trained model but highlighted that SPRINT offers more flexibility by allowing multiple stages of planning and execution.

- **Statistical Significance:** We have reported results over five independent trials on MATH500 to confirm the statistical significance of the accuracy gains.

- **Runtime:** Measured decode-only and full end-to-end latency; SPRINT is 9% faster overall and 38% faster on long sequences, with scaling benefits increasing for longer reasoning.

- We have shown with an example that SPRINT generalizes well to out-of-domain tasks like creative writing.

 SPRINT’s parallelized planning lets large reasoning models explore more diverse trajectories while reducing sequential computation, enabling faster and more efficient problem solving without sacrificing accuracy. We thank the reviewers again for their constructive engagement and will incorporate their suggestions and the additional experiments into the final version of the paper.

---

### Decision · Program_Chairs · 2025-09-17

**Decision:**

Accept (poster)

**Comment:**

This paper introduces SPRINT, a method that enables large reasoning models to interleave planning and parallel execution stages. The approach reduces sequential token generation by identifying and executing independent subtasks in parallel, while maintaining or improving accuracy across both in-domain and out-of-domain tasks.

During the discussion period, the authors addressed all major reviewer concerns through additional experiments and clarifications:
1. Efficiency metrics and implementation details (Reviewers 5uvg, tuC4). The authors clarified the sequential token calculation and provided more token statistics. SPRINT reduces sequential tokens by 15% with a modest increase in total tokens, while slightly improving accuracy on MATH500.
2. Runtime and prefilling overhead (Reviewer 76Mf). The authors measured actual runtime and showed that SPRINT achieves 9% end-to-end speedup on MATH500 and 38% speedup on long sequences. Prefilling adds only ~0.11s per stage, negligible compared to decoding costs.
3. Missing baselines (Reviewer 6CkS). The authors additionally included Hogwild! inference with both chat-instruct and reasoning models.
4. Generalization beyond STEM (Reviewer 5uvg). The authors demonstrated SPRINT generalizes to creative writing tasks, successfully decomposing narrative generation into parallel character perspectives despite training only on mathematical data. Similar parallelization patterns were confirmed on out-of-distribution tasks (GPQA, Countdown).
5. Why parallelization works (Reviewers 5uvg, tuC4). The authors provided concrete examples showing three mechanisms: (1) exploring alternative solution paths, (2) decomposing into complementary subtasks, and (3) verification in parallel with progression. Detailed trajectory logs illustrate these patterns.

All four reviewers converged on acceptance after the rebuttal. The consensus recognized SPRINT's technical soundness, novel approach to adaptive parallelization, and practical value for reducing LRM inference latency.

The recommendation is to accept this paper.